# Unassisted selective solar hydrogen peroxide production by an oxidised buckypaper-integrated perovskite photocathode

Rashmi Mehrotra [1,2,4], Dongrak Oh [1,2,4] & Ji-Wook Jang [1,2,3✉]

Hydrogen peroxide ($H_2O_2$) is an eco-friendly oxidant and a promising energy source possessing comparable energy density to that of compressed $H_2$. The current $H_2O_2$ production strategies mostly depend on the anthraquinone oxidation process, which requires significant energy and numerous organic chemicals. Photocatalyst-based solar $H_2O_2$ production comprises single-step $O_2$ reduction to $H_2O_2$, which is a simple and eco-friendly method. However, the solar-to-$H_2O_2$ conversion efficiency is limited by the low performance of the inorganic semiconductor-based photoelectrodes and low selectivity and stability of the $H_2O_2$ production electrocatalyst. Herein, we demonstrate unassisted solar $H_2O_2$ production using an oxidised buckypaper as the $H_2O_2$ electrocatalyst combined with a high-performance inorganic-organic hybrid (perovskite) photocathode, without the need for additional bias or sacrificial agents. This integrated photoelectrode system shows 100% selectivity toward $H_2O_2$ and a solar-to-chemical conversion efficiency of ~1.463%.

---

[1] School of Energy and Chemical Engineering, Ulsan National Institute of Science and Technology (UNIST), Ulsan 44919, Republic of Korea. [2] Department of Energy Engineering, Ulsan National Institute of Science and Technology (UNIST), Ulsan 44919, Republic of Korea. [3] Emergent Hydrogen Technology R&D Centre, Ulsan National Institute of Science and Technology (UNIST), Ulsan 44919, Republic of Korea. [4] These authors contributed equally: Rashmi Mehrotra, Dongrak Oh. ✉email: jiwjang@unist.ac.kr

Hydrogen peroxide ($H_2O_2$) is an eco-friendly oxidising agent that only produces $H_2O$ and $O_2$ as by-products and has been applied in wastewater treatment, the paper industry, and pulp bleaching[1,2,3]. Moreover, the energy density of aqueous $H_2O_2$ (60%; $3.0 MJ L^{-1}$) is equivalent to that of compressed $H_2$ ($2.8 MJ L^{-1}$ at 35 MPa)[1], while its aqueous phase facilitates its storage and transfer[3]. Currently, more than 95% $H_2O_2$ is commercially manufactured by the "anthraquinone oxidation" process, which consists of multi-step reaction processes and many separation steps that require significant energy and numerous organic chemicals[4,5]. In addition, precious Pd-based catalysts and high-pressure $H_2$ used during the hydrogenation reaction increase the cost of the process and introduce many safety issues[6–8].

In this regard, solar $H_2O_2$ production via the single-step reduction of $O_2$ using a photocatalyst offers a simple, safe, and sustainable alternative[9–11]. To date, several strategies have been reported for efficient solar $H_2O_2$ production[10,11], wherein the main focus lies in increasing the solar-to-chemical ($H_2O_2$) conversion efficiency (SCC; %) by improving the photoelectrode performance and electrocatalyst selectivity. Current solar $H_2O_2$ production applications only utilise inorganic-based photoelectrodes. Such electrodes intrinsically possess poor charge transfer characteristics with non-tuneable band gaps and band positions and thus exhibit low performances[12–16]. With respect to $H_2O_2$ electrocatalysts, porphyrin-based materials containing 3d transition metals have been studied and applied because of their high selectivities for $H_2O_2$ production; however, these metals are expensive and exhibit low stability, owing to the easy cleavage of the central metal (Co) and N bonds[17,18].

Herein, we introduce a highly efficient inorganic–organic methylammonium lead triiodide ($MAPbI_3$ perovskite, PSK)-based photocathode, which is passivated by Field's metal (FM) with the selective and stable oxidised buckypaper (O-BP) electrocatalyst for $H_2O_2$ production (Fig. 1). PSK materials offer many advantages over inorganic-based photoelectrodes including a longer charge carrier diffusion length[19], bandgap tunability (composition-dependent)[20], and an enhanced light absorption coefficient[21,22]. However, despite these superior properties, these materials have not attracted much attention, owing to their low stability in aqueous solution[23]. With this in mind, in this work, PSK was efficiently protected[24] by an FM-connected O-BP electrocatalyst. The O-BP was fabricated by the filtration of oxidised carbon nanotubes (O-CNTs), which are a known, efficient $H_2O_2$ electrocatalyst[25]. The cross-linked O-CNTs structure in the O-BP acts as a selective and efficient electrocatalyst (Faradaic efficiency, FE = 100%). FM (melts at ~65 °C and solidifies at ambient temperature)[26] is used to tightly attach the PSK film to the O-BP, but it also facilitates charge transfer between them and acted as a protection layer for the PSK film from water penetration. This integrated system shows good stability over 45 h compared to that of the non-passivated PSK layer, which loses its activity within a few minutes[27,28]. Moreover, a $H_2O_2$ production rate of ~0.751 $\mu mol min^{-1} cm^{-2}$ and an SCC of ~1.463% were achieved[29].

## Results

### Characterisation and $H_2O_2$ production performance of carbon materials

O-CNTs were synthesised by the acid treatment of CNTs at 80 °C for 48 h in nitric acid (60 wt%)[25]. Buckypaper (BP) and O-BP were synthesised by the filtration of CNTs and O-CNTs, respectively (see Methods section). The scanning electron micrographs (SEM) and images (inset) of BP and O-BP (Supplementary Figs. 1a and 2a, respectively) show the randomly self-aligned CNTs and O-CNTs, which are closely linked to each other. As shown in Supplementary Fig. 1b and Fig. 2b, BP and O-BP are densely stacked but possess high surface areas with many mesopores (Supplementary Figs. 2 and 3). These results indicate that BP and O-BP have similar physical properties. Next, Raman and X-ray photoelectron spectroscopy (XPS) analyses were performed to investigate the chemical structures of the two materials. The Raman spectra (Fig. 2c and Supplementary Fig. 4) display $D$ (~1350 $cm^{-1}$)-to-$G$ (~1580 $cm^{-1}$) peak ratios of 1.08 and 1.14 for BP and O-BP, respectively, indicating that the acidic treatment of CNTs causes structural disorder, thereby producing

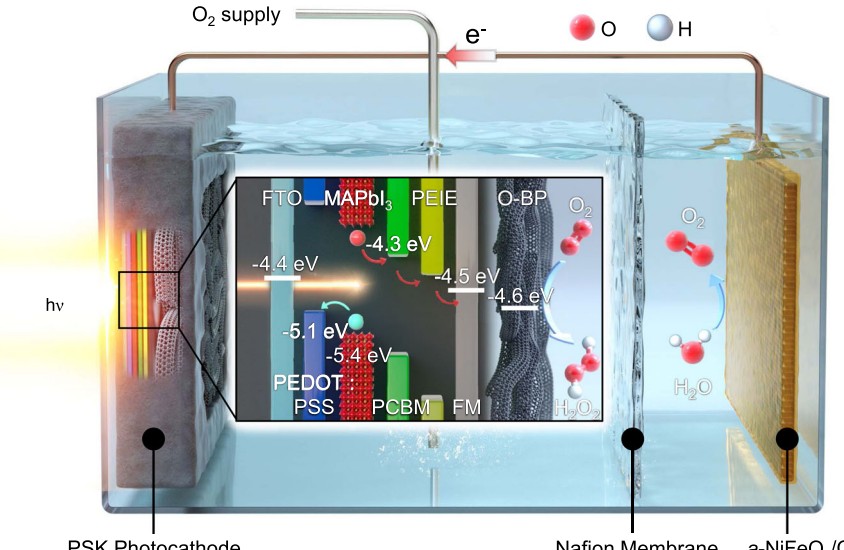

**Fig. 1 Schematic illustration of the system demonstrating unassisted solar hydrogen peroxide ($H_2O_2$) production.** An inverse-structure methylammonium lead triiodide ($MAPbI_3$) perovskite (PSK) photocathode with oxidised buckypaper (O-BP) and an amorphous-based nickel–iron oxide (*a*-NiFeO$_x$) with a carbon paper (CP) substrate are connected for bias-free $H_2O_2$ production. Embedded copper wires connect the photocathode and anode to enable the photogenerated hole–electron pairs to flow in a closed circuit. Dissolved $O_2$ is reduced to $H_2O_2$ by photocathode-mounted O-BP. The oxygen evolution reaction (OER) occurs at the *a*-NiFeO$_x$/CP. Definitions: PCBM, [6,6]-phenyl C$_{61}$ butyric acid methyl ester; PEIE, polyethyleneimine; and FTO, fluorine-doped tin oxide. The red-coloured arrow shows the direction of the movement of the electrons in the illustration while the simultaneously generated holes' direction is depicted by the blue-coloured arrow.

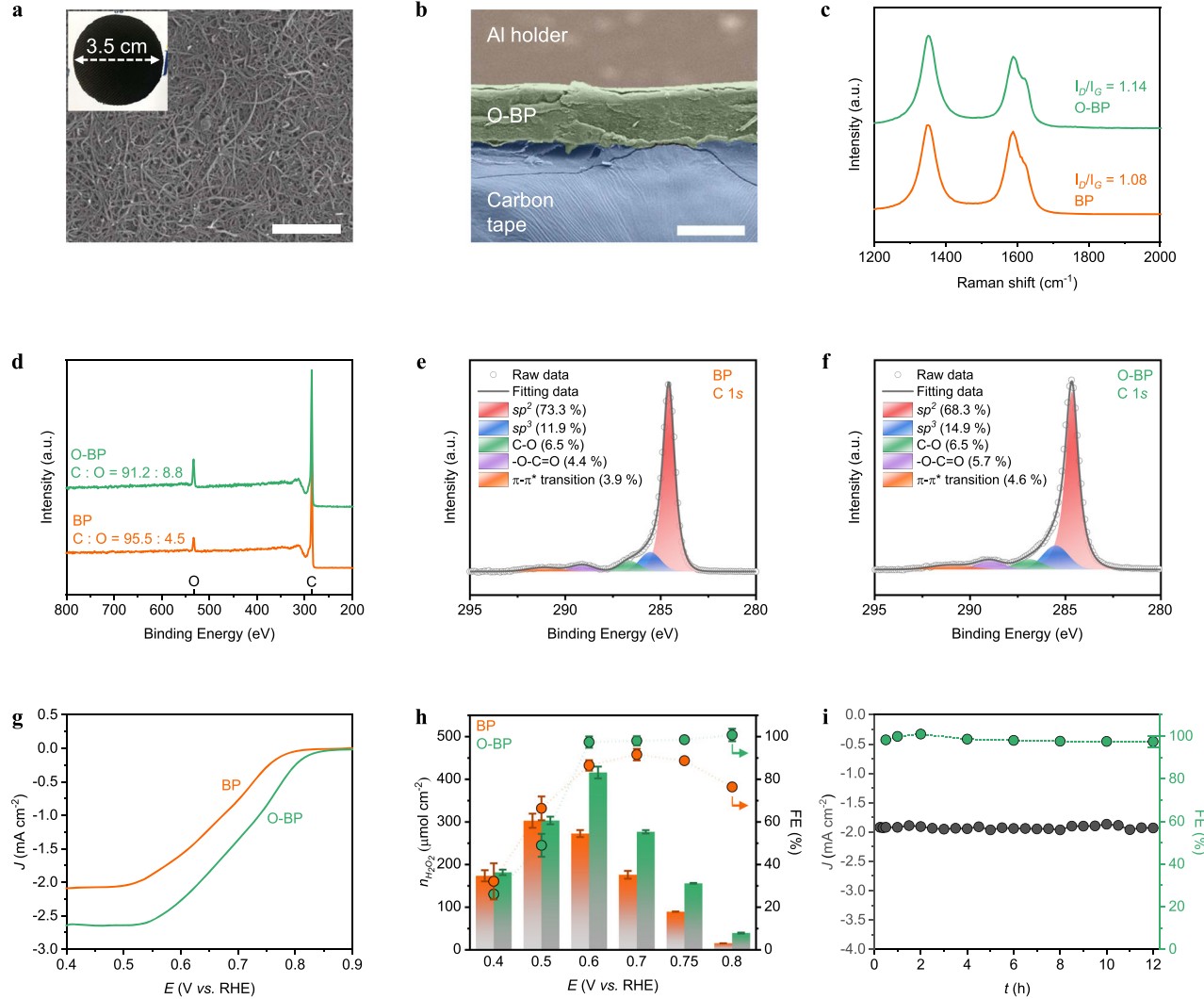

**Fig. 2 Characterisation and oxygen reduction performance of carbon materials. a** Top-view scanning electron micrographs of O-BP. Inset photograph, O-BP after being removed from a polytetrafluoroethylene membrane. Scale bar, 1 μm. **b** Cross-section scanning electron micrographs of O-BP. Scale bar, 100 μm. **c** Raman spectra of BP and O-BP. **d** X-ray photoelectron spectroscopy (XPS) surveys of BP and O-BP. Deconvoluted carbon 1s spectra of BP (**e**) and O-BP (**f**). **g** Capacitive current compensated linear sweep voltammetry (LSV) of BP and O-BP evaluated at a 10 mV s$^{-1}$ scan rate. **h** Amount of H$_2$O$_2$ produced ($n$) and the faradaic efficiency (FE) of BP and O-BP during 12 h of controlled potential electrolysis (CPE). Here, the error bars indicate the standard deviation. **i** Long-term CPE test of O-BP at −0.6 V vs. RHE. **g–i** All experiments were conducted in 0.1 M KOH solution (pH ~13.17) under O$_2$ and ambient conditions with stirring.

many defects on O-CNTs[25]. This is also observed in the transmission electron microscopy (TEM) images (Supplementary Fig. 5).

To further investigate their detailed surface structures, XPS analysis was also conducted (Fig. 2d–f and Supplementary Fig. 6). Figure 2d shows that compared to those of the BP samples, the oxygen-to-carbon ratio increased for O-BP, indicating that the CNTs are oxidised during the acid treatment. The oxygen peaks that appear for BP are due to surface contamination with oxygen-containing species, which are likely adsorbed during the filtration process of the CNT-containing solution[30–33]. Thermogravimetric (TGA) analysis (Supplementary Fig. 7) showed that there was no weight loss in the non-functionalised carbon materials, including the BP and CNTs, while a weight loss of about 10% was observed for the functionalised (O-BP and O-CNTs) carbon materials due to detachment of functional group; this weight loss occurred when the temperature was increased up to 500 °C. This result indicates that the oxygen peak observed for BP originates from oxygen-containing impurities rather than functional groups, and

its absolute amount is very low. Contact angle measurements of BP and O-BP also support this (Supplementary Fig. 8). The carbon 1s spectra of BP (and O-BP) were deconvoluted into five contributions with $sp^2$ carbon, $sp^3$ carbon (defect sites), C–O, and –COOH peaks at 284.6 eV (and 284.7 eV), 285.5 eV (and 285.4 eV), 286.4 eV (and 286.7 eV), and 289.1 eV (and 289.1 eV), respectively; and the characteristic shakeup line of carbon was noted at 291.1 eV (and 291.1 eV; Fig. 2e, f). Here, the shakeup line occurs due to the instant energy loss of the emitted photoelectrons following the excitation of the ground-level electrons (π–π* transition)[34]. These XPS results show that O-BP possesses more defect sites than BP, including functional groups such as O-adhering carbons and structural disorders such as $sp^3$ carbons that are known as highly active H$_2$O$_2$ production sites for the two-electron oxygen reduction reaction (ORR) pathway[35–38]. In addition, owing to their increased amounts of O-adhering carbons, the O-CNTs were more dispersible in deionised water and *N,N*-dimethylformamide (DMF), indicating that O-BP should exhibit a more intimate interaction with water than BP

(Supplementary Fig. 8a–c). This was also verified by contact angle measurements in which the contact angles of O-BP and BP were 25° and 134°, respectively, indicating that O-CNTs are more hydrophilic than CNTs (Supplementary Fig. 8d, e). The O-CNTs also displayed a more negative zeta-potential in aqueous 0.1 M KOH than the CNTs, indicating better $H_2O_2$ ($HO_2^-$ in alkaline solution) generation with more facile product detachment ($O_2 + H_2O + 2e^- \rightarrow HO_2^- + OH^-$; Supplementary Fig. 8b)[39]; this is likely due to the higher repulsive interactions between the negatively charged functional groups on the O-CNTs and the negatively charged $HO_2^-$ species in aqueous 0.1 M KOH (CNTs have less functional groups than O-CNTs).

To evaluate the ORR performance of BP and O-BP, capacitive compensated linear sweep voltammetry (LSV) curves of BP and O-BP were first obtained in an aqueous solution (0.1 M KOH, pH ~13.17) under continuous $O_2$ flow (50 cc $min^{-1}$; Fig. 2g and Supplementary Fig. 9a, b). As expected, O-BP shows a higher current density and more positive onset potential than those of BP (Fig. 2g). The onset potential of O-BP was 0.815 V vs. RHE, which is higher than the standard reduction potential ($E° = 0.74$ V vs. RHE; possible initially by the Nernst equation), indicating the presence of negligible overpotentials of O-BP for two-electron ORR (Fig. 2g). Here, the onset potential is defined as the potential at 0.1 mA $cm^{-2}$. Moreover, the selectivity of O-BP toward $H_2O_2$ production approached 100%, which is higher than that observed for BP at potentials above 0.6 V (Fig. 2h). When the applied over-potential was increased to 0.5 V and 0.4 V vs. RHE, the selectivities of both BP and O-BP decreased (Fig. 2h); this was because the degradation rate of $H_2O_2$ increased due to the faster accumulation rate of $H_2O_2$ at these high negative potentials (Supplementary Fig. 9c, d and Supplementary Fig. 10)[40]. Notably, O-BP produced higher absolute amounts of $H_2O_2$ than did BP at all the applied potentials (Fig. 2h). As denoted previously, acid treatment induced the oxidation of the CNT surface, resulting in many functional groups with disordered defect sites, which enhanced both the activity and selectivity of the ORR to $H_2O_2$. The current density and FE were stable for 12 h with 100% selectivity at 0.6 V vs. RHE, where O-BP displayed the highest selectivity and $H_2O_2$ amount (Fig. 2i). The thickness of O-BP did not affect its performance or stability if the value was greater than ~100 μm, as shown in Supplementary Fig. 11 and Supplementary Fig. 12. Furthermore, the O-BP electrocatalyst was stable for 100 h with the renewal of fresh electrolyte periodically (Supplementary Fig. 13).

**Characterisation and ORR activity of O-BP/FM/PSK photocathode.** PSK inverted structure (p-i-n) thin films were fabricated by the sequential deposition of poly(3, 4-ethylenedioxythiophene) polystyrene sulfonate (PEDOT:PSS) as the hole transfer layer (HTL), a $MAPbI_3$ perovskite layer (as a photo-absorber), phenyl-$C_{61}$-butyric acid methyl ester (PCBM) as the electron transfer layer (ETL), and polyethyleneimine (PEIE) (as the charge collector layer)[41–44] on fluorine-doped tin oxide (FTO) substrate, using all solution-based methods. Notably, the designed system does not use an expensive Au or Ag layer as the counter electrode[45,46]. In this study, the $MAPbI_3$ PSK layer was deposited by modified two-step methods.

The top-view and cross-sectional scanning electron micrographs in Supplementary Fig. 14 demonstrate mesoporous lead iodide ($PbI_2$) structure formation during spin-coating due to the hot-casting of $PbI_2$ precursors, including dimethyl sulfoxide (DMSO) additive as a high boiling point solvent (creates pores during evaporation)[47,48]. Figure 3a, b shows top-view and cross-sectional scanning electron micrographs of $MAPbI_3$ while Supplementary Fig. 15 shows the PSK layer formed from the

intercalation of methylammonium iodide (MAI) organic cations into the $PbI_2$ voids, which allows the faster conversion of the yellow $PbI_2$ to black $MAPbI_3$ PSK films. UV-vis spectroscopy revealed enhanced light absorption with a broad absorption band at ~775 nm in the entire visible range (400–850 nm; Supplementary Fig. 16a). The bandgap determined from the Kubelka–Munk equation was 1.57 eV, which approached the theoretical value for $MAPbI_3$ PSK (Supplementary Fig. 16b)[49]. Next, we passivated the PSK layer with FM followed by the O-BP electrocatalyst to prevent water penetration, as denoted previously. We attached the PSK layer to O-BP using solid FM at temperatures above its melting point of 65 °C. Thus, at ambient temperature, its phase changed to a solid, creating a tight junction between the PSK film and O-BP that resulted in intimate charge transfer between the layers.

The photoelectrochemical (PEC) performance for $H_2O_2$ production when using the integrated O-BP/FM/PSK photocathode was then investigated under simulated 1-sun irradiation and continuous $O_2$ flow. We optimised the thicknesses of the individual layers, including the HTL, PSK photo-absorber, and ETL, to enhance the performance of the O-BP/FM/PSK photocathode, as shown in Supplementary Fig. 17 and Supplementary Table 1. In addition, we found that the basic conditions (0.1 M KOH, pH ~13) with an optimised pH provided better performance than those of the neutral (0.05 M $Na_2SO_4$, pH ~ 7) and acidic (0.05 M $H_2SO_4$, pH ~ 1) conditions (Supplementary Fig. 18). Figure 3c shows the LSV scans of the O-BP electrocatalyst (green) and FM/PSK photocathode (grey), and the cyclic voltammograms (CV) and averaged CV (avg. CV) scans of the O-BP/FM/PSK photocathode device (blue). The photocurrent density and onset potential exhibited by the optimised integrated device were much higher ($-11.15$ mA $cm^{-2}$ at 0 V vs. RHE and 1.77 V vs. RHE, respectively) than those of FM/PSK ($-3.18$ mA $cm^{-2}$ at 0 V vs. RHE and 1.2 V vs. RHE, respectively) in 0.1 M KOH. Furthermore, the small curvature at ~1.5 V vs. RHE (accounting for ~0.27 V of overpotential) was attributed to the limited solubility of $O_2$ in $H_2O$ (as was electrocatalyst; Supplementary Fig. 9b). Meanwhile, the photocurrent density continuously reaches the maximum value at more negative potential range owing to (i) the experimental capacitance (practically CV for non-faradaic reaction is not rectangular in shape) and (ii) the hydrogen evolution reaction. The early onset potential of 1.77 V vs. RHE obtained from the avg. CV scan of the O-BP/FM/PSK photocathode device mitigated the route for $H_2O_2$ generation using only a single photo-absorber and, to the best of our knowledge, this value is one of the most positive values[39].

The O-BP/PSK photocathode without use of FM connecting O-BP and PSK layers exhibits a very low photocurrent density of 110 μA $cm^{-2}$ at 0 V vs. RHE and lost its photoactivity within 30 min (Supplementary Fig. 19). In addition, less than 12% of the photocurrent of the FM/PSK photoelectrode remained after 10 h at 0.6 V **vs.** RHE (Supplementary Fig. 20). These results show the importance of FM and O-BP in forming an intimate junction between the PSK layer and O-BP to stabilise the PSK layer. We then compared the performances of the O-BP (sheet) /FM electrocatalyst and an O-CNTs (powder)/FM electrocatalyst to investigate if a sheet or a powder would more suitably combine with FM to give an electrocatalyst with better performance and stability (Supplementary Fig. 21). There was a negative shift in the onset potential of O-CNTs/FM of 170 mV relative to that of O-BP/FM (Supplementary Fig. 21a). The initial current density of O-CNTs/FM at 0.6 V vs. RHE in 0.1 M KOH ($O_2$ purging) was only $-0.5$ mA $cm^{-2}$, which was approximately four times less than that of O-BP/FM. Furthermore, the O-CNTs/FM electrocatalyst lost its activity within a few minutes due to the detachment of the O-CNTs powder from the FM; contrastingly,

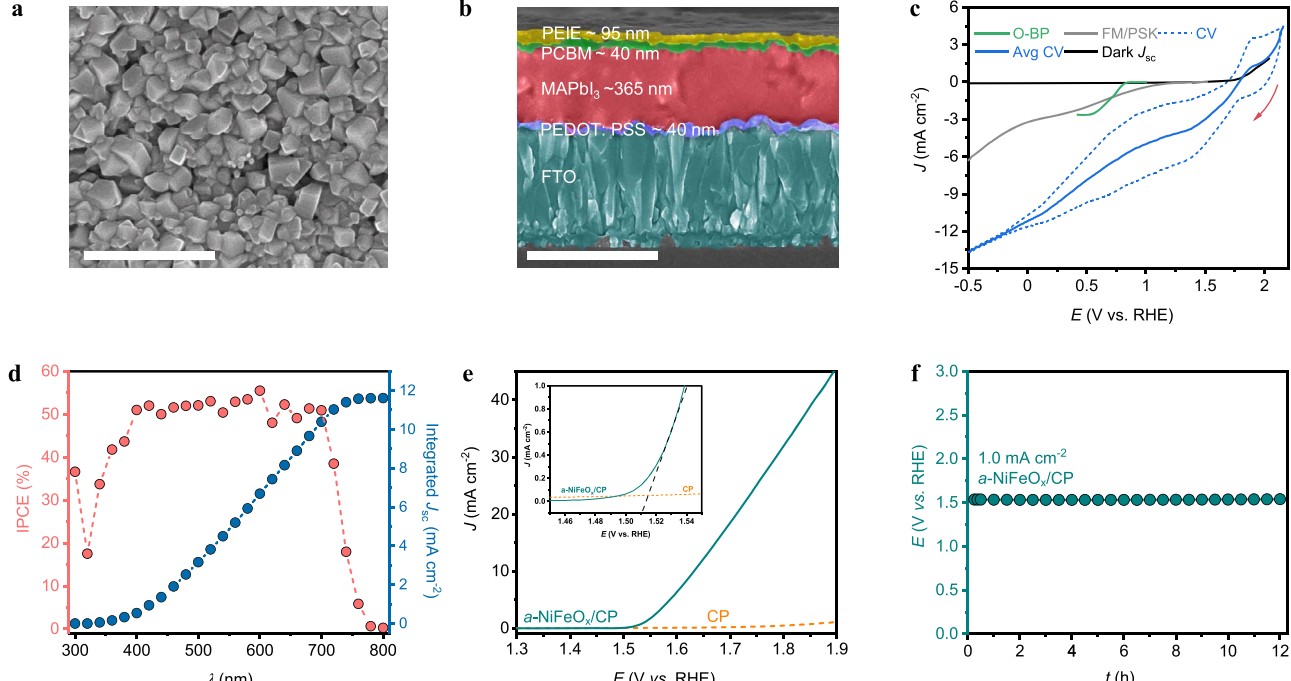

**Fig. 3 Characterisation and ORR activity of the O-BP-integrated PSK photocathode. a**, **b** Top-view and cross-sectional scanning electron micrographs of the formed MAPbI$_3$ PSK showing the thickness of the individual layers. **c** $J$–$E$ responses of the O-BP electrocatalyst, Field's metal (FM)/PSK photocathode, and integrated O-BP/FM/PSK photocathode device, exhibiting an anodic shift. Integrated O-BP/FM/PSK photocathode devices were simulated under 1-sun and an air mass of 1.5 G, with the averaged cyclic voltammograms (Avg. CV) scans under light and no irradiation (red arrow indicates the scan direction during the experiment) in the presence of an O$_2$ supply. **d** Wavelength-dependent spectral response (i.e., incident-photon-to-current efficiency (IPCE)) and the corresponding integrated photocurrent density of the integrated device at 0 V vs. RHE. **e** $J$–$E$ response of the $a$-NiFeO$_x$/CP anode without stirring. Inset depicts the exact overpotential for the OER. **f** Chronopotentiometry test of the $a$-NiFeO$_x$/CP anode conducted in a three-electrode set-up. **c**–**f** Experiments conducted in a 0.1 M KOH solution (pH ~13.17) under ambient conditions with a scan rate of 5 mV s$^{-1}$. All scale bars equal 1 μm.

O-BP/FM exhibited stable performance (Supplementary Fig. 21b). These results indicate that a sheet forms a more intimate and stable junction with FM than a powder, making the former more suitable for use in a H$_2$O$_2$ electrocatalyst.

To further access the performance of the integrated O-BP/FM/PSK device, we measured the incident-photon-to-current efficiency, IPCE (%), as a function of the wavelength in the three-electrode configuration against Pt-mesh as the counter electrode and Hg/HgO as the reference electrode in 0.1 M KOH (~pH 13.17) solution. The IPCE was calculated as follows (Eq. (1)):

$$IPCE(\%) = \left( \frac{1240 \times [J_P]}{[P_\lambda] \times \lambda} \right) \times 100 \tag{1}$$

with J$_P$ as the photocurrent density (μA cm$^{-2}$), P$_\lambda$ as the power density at a particular wavelength λ, and λ as wavelength of the incident light (nm). We obtained an IPCE of ~52% in the range 400–700 nm, corresponding to 11.61 mA cm$^{-2}$ as the integrated photocurrent density (Fig. 3d). In addition, the IPCE showed an onset in the photocurrent density at 775 nm, which well-matched the PSK absorption band (Supplementary Fig. 16a).

On the anodic part, amorphous-based nickel–iron oxide ($a$-NiFeO$_x$)/carbon paper (CP) was used as an oxygen evolution reaction (OER) electrocatalyst, presenting an onset potential of ~1.52 V vs. RHE (achieved current density, ~1 mA cm$^{-2}$; inset of Fig. 3e) and good stability for 12 h (Fig. 3f). Moreover, there were no significant changes in the LSV curves and X-ray diffraction (XRD) data before and after the reaction, further verifying the stability of the $a$-NiFeO$_x$ during the reaction (Supplementary Figs. 22 and 23).

**Unassisted H$_2$O$_2$ production with integrated photocathode.** Unassisted solar H$_2$O$_2$ production, where an additional bias or sacrificial agent is not required, is deemed optimal. To demonstrate this, we combined the O-BP/FM/PSK photocathode with the $a$-NiFeO$_x$/CP anode in two ways; namely, in a two-electrode configuration (applied potential equal to 0 V) and by connecting the two electrodes with Cu wire. The detailed configuration of the working mechanism for the solar H$_2$O$_2$ production is shown in Fig. 1. The light incident on the FTO-side of our integrated O-BP/FM/PSK photocathode is absorbed by the MAPbI$_3$ absorber, thereby generating electron–hole pairs. The photogenerated electrons reach the electrocatalytic surface of the O-BP through the ETL and FM and the photogenerated holes reach the $a$-NiFeO$_x$/CP anode, for the simultaneous production of H$_2$O$_2$ and O$_2$, respectively. The arrows shown in Fig. 1 demonstrate the movement of the photogenerated holes (blue) and electrons (red). The employed two-compartment reactor comprised a Nafion membrane (Supplementary Fig. 24a), which separated the anolyte and catholyte while preventing the re-oxidation of H$_2$O$_2$. Figure 4a shows the overlapped LSV scans of the O-BP/FM/PSK photocathode (obtained from the avg. CV scan, as denoted previously) and $a$-NiFeO$_x$/CP anode in a 0.1 M KOH electrolyte (pH ~13.17) with constant O$_2$ flow and stirring. An operating current of ~2.51 mA cm$^{-2}$ at 1.56 V vs. RHE was obtained with LSV graph of anode and photocathode itself. This demonstrated the probability of generating PEC H$_2$O$_2$ in a bias-free mode (without applying any voltage) by only deploying a single photo-absorber. Figure 4b reveals that this system continuously produced H$_2$O$_2$ with 100% selectivity. Moreover, the performance of the O-BP/FM/PSK photocathode did not show any sign of decrease during the 45 h of stability tests, as shown in Supplementary Fig. 25.

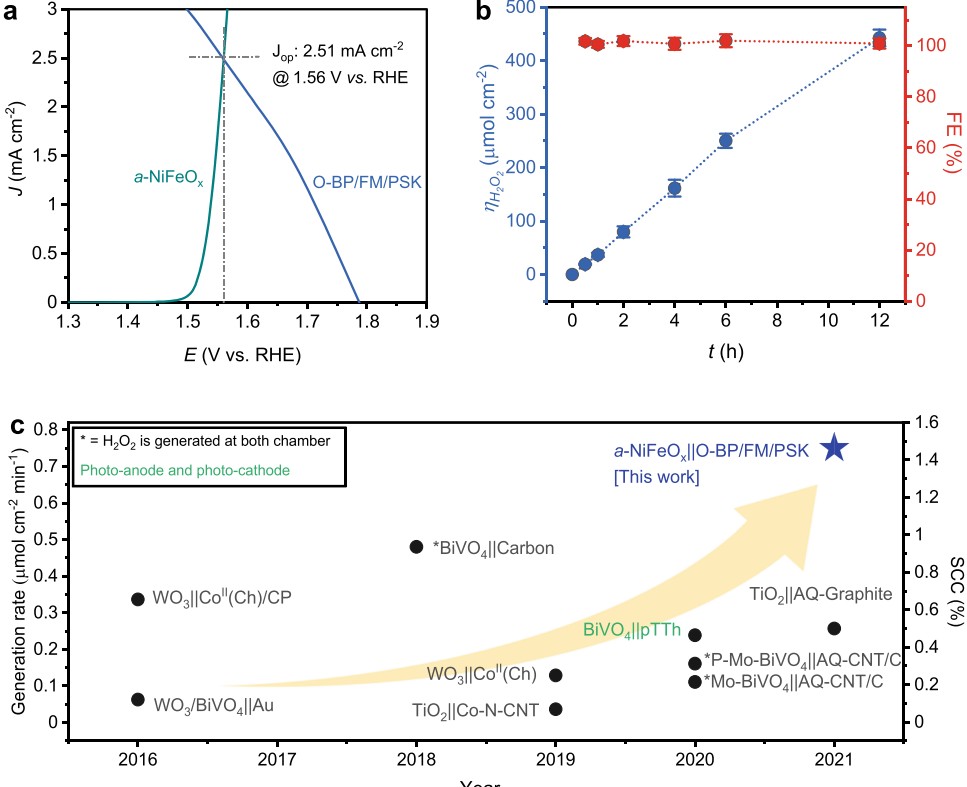

**Fig. 4 Unassisted solar $H_2O_2$ production with integrated photocathode system. a** Operating point, obtained from the overlap of the $a$-NiFeO$_x$/CP anode and the integrated system of the O-BP/FM/PSK photocathode, for the generated solar $H_2O_2$ affords a photocurrent of ~2.51 mA cm$^{-2}$ for generating solar $H_2O_2$ in the catholyte. **b** In a two-electrode set-up under simulated 1-sun conditions, the unassisted $H_2O_2$ production ($n$) of our integrated O-BP/FM/PSK device could continuously produce ~442 ± 15.2 μmol cm$^{-2}$ of $H_2O_2$ for 12 h, without any degradation, at 0 V. Here, the error bars indicate the standard deviation based on averaged data of five integrated devices illustrating $n_{H2O2}$ and FE (%). **c** Graph showing the generated rate of solar $H_2O_2$ on the left $y$-axis, while the corresponding solar-to-chemical conversion efficiency (SCC;%) has been marked on the right $y$-axis (Supplementary Table 2). The highest SCC efficiency peaked after 2 h of reaction, also demonstrated in Supplementary Fig. 28.

XRD, SEM, and XPS analyses of the O-BP/FM/PSK photocathode after the stability tests (Supplementary Fig. 26) showed no significant changes in its crystal structure, morphology, or electronic distribution. When we directly connected the O-BP/FM/PSK photocathode with the $a$-NiFeO$_x$/CP anode via a copper wire, continuous and stable $H_2O_2$ production was observed for 12 h with near-similar amounts of $H_2O_2$ (407 ± 8.5 μmol cm$^{-2}$; Supplementary Fig. 27) to those attained with the two-electrode system (442 ± 15 μmol cm$^{-2}$; Fig. 4b). The above results indicate that unassisted solar $H_2O_2$ production was successfully achieved using our designed high-performance and stable O-BP/FM/PSK photocathode system. Moreover, we recorded an SCC of ~1.463% after 2 h (average SCC of 1.35 ± 0.073% after 6 h) (Fig. 4c, Supplementary Fig. 28, and Supplementary Table 2). The SCC (%) was calculated as follows (Eq. (2)):

$$SCC(\%) = \left( \frac{[\triangle G°] \times [Produced\ amount\ of\ H_2O_2]}{[Pin] \times [illuminated\ area]} \right) \times 100 \quad (2)$$

with $\Delta G°$ as Gibbs free energy gain due to $H_2O_2$ formation (117 KJ mol$^{-1}$), the produced amount of $H_2O_2$ as concentration of the generated $H_2O_2$ × volume of the catholyte (7.5 mL), $P_{in}$ as power density of the incident solar light, and illuminated area as 0.25 cm$^2$ of the integrated device used in the $H_2O_2$ producing system.

## Discussion

In summary, we successfully passivated a high-performance PSK photoelectrode using O-BP, which acts as a selective electrocatalyst for $H_2O_2$ production and a protection layer when

combined with FM to prevent water penetration. The $H_2O_2$ production selectivity of the O-BP electrocatalyst approached 100%, with good stability for 100 h with the reaction system being periodically supplied with fresh electrolyte. In addition, we demonstrated unassisted solar $H_2O_2$ production using a two-electrode system comprising a O-BP/FM/PSK photocathode and an $a$-NiFeO$_x$ anode that demonstrated SCC of ~1.463%. Furthermore, this system was stable for 45 h under 1-sun illumination with 100% $H_2O_2$ production selectivity. The SCC (%) and stability of the overall system can be further improved by optimising the PSK, ETL, HTL, and electrocatalyst components. Moreover, this passivation strategy can be applied to various reactions, such as $CO_2$ and $N_2$ reduction, by simply changing the electrocatalyst on the PSK photoelectrode; thus, this work provides a protocol to efficiently produce other valuable solar chemicals for carbon-based fuel applications and $NH_3$.

## Methods

**Preparation of oxidised CNTs (O-CNTs)**. Multiwalled CNTs (MWCNTs) were oxidised by a well-known acid treatment[25]. First, the MWCNTs (400 mg) and nitric acid (60 wt%, 400 mL) were added to a 500 mL three-necked round-bottomed glass flask. Next, the flask was equipped with a magnetic stirrer and thermometer and placed in a thermostat within the reflux system. The temperature was maintained at 80 °C for 48 h, after which the slurry was cooled to room temperature (at ~25 °C), poured out, filtered, and washed with distilled water several times until a neural pH was attained. Finally, the sample was dried overnight at 60 °C in a vacuum oven.

**Carbon material sheet preparation**. Carbon powder (exactly 50 mg) and 150 mL of DMF (150 mL; Sigma–Aldrich, anhydrous, 99%) were placed in a 250 mL

chemical storage bottle and dispersed by tip sonication with a 25% amplitude for 15 min followed by bath ultrasonication for 2 h. After sonication, the dispersion was filtered through a polytetrafluoroethylene membrane (0.45 µm pore size) and washed with acetone (600 mL) in three instalments. Next, the as-formed carbon sheet was carefully removed from the membrane and dried overnight in a vacuum oven at 60 °C.

**Electrocatalyst electrode preparation**. To prepare the electrodes, a carbon sheet (~1.0 × 0.7 cm$^2$) was cut and placed on the FM-covered glass, interfaced with insulated copper wire and silver conductive paint, and covered by epoxy resin to form a geometrical area of 0.25 cm$^2$ (Supplementary Fig. 24b).

**Perovskite photocathode fabrication**. A PbI$_2$ (Sigma–Aldrich, ~1 M, > 99% purity) precursor was formed by dissolving PbI$_2$ powder (exactly 462 mg) in a 9:1 v:v DMF and DMSO (Sigma Aldrich, anhydrous, 99%). A solution (20 mg mL$^{-1}$) of PCBM in chlorobenzene (Sigma–Aldrich, anhydrous, 99%) was maintained under constant stirring at 70 °C overnight, while MAI (10 mg mL$^{-1}$; a great solar cell) was dissolved in 2-propanol (Sigma Aldrich, anhydrous, 99%). The inverse structure of the mixed halide perovskite films was prepared via a previously reported two-step spin-coating method[50]. Briefly, PEDOT:PSS was spin-coated at 4000 rpm for 60 s onto a 1 × 1.5 cm$^2$ FTO glass substrate and annealed at 120 °C for 20 min to form the HTL. The relative humidity was maintained at ~20%. The samples were then immediately transferred to a N$_2$-filled glove box. Samples were preheated at 146 ± 4 °C on a hot plate for around 5 min each, spin-coated at 2200 rpm for 20 s with an acceleration of 440 rpm s$^{-1}$ by dispensing 150 µL hot PbI$_2$ ink (150 µL), and then placed in a covered petri-dish for 7 min. The films were then annealed at 70 °C for 15 min and subsequently cooled to glove box temperature (~27–29 °C). Dark brown MAPbI$_3$ films were developed by dispensing MAI (0.063 M, 350 µL) at 0 rpm for 40 s and 2700 rpm for 20 s to wash off the remnant propanol from the FTO. After spin coating, the films were annealed at 85 °C for 10 min. Subsequently, the PCBM–chlorobenzene solution (80 µL) was deposited at 2000 rpm for 30 s to give the completed structure. For better extraction of the electrons by the catalyst, a thin layer of PEIE solution (0.1 mL in 5 mL of 2-propanol) was spin-coated at 3000 rpm for 30 s. To protect the perovskites from any degradation, an FM sheet (1.0 × 0.7 cm$^2$) was melted onto the perovskite layer at temperatures ≤70 °C and kept inside the glove box overnight to dry.

**Perovskite device integration**. To prepare the photocathodes, O-BP sheet (~1.0 × 0.7 cm$^2$) was interfaced precisely on the melted FM/PSK photocathode, which was covered simultaneously with an epoxy resin to provide a geometrical area of 0.25 cm$^2$ on the electrocatalyst and back-side of the FTO substrate (Supplementary Fig. 24c). Finally, an insulated copper wire was connected to complete the device fabrication.

**a-NiFeO$_x$/CP anode**. The anode was prepared according to a previously reported method[51,52]. Briefly, an a-NiFeO$_x$ solution (1 mL) was prepared by weighing iron (III) ethylhexanoate (0.0358 g, 50% w/w in mineral spirits, Strem Chemicals) and nickel (II) ethylhexanoate (0.0169 g, 78% w/w in 2-ethylhexanoic acid, Strem Chemicals) in a vial and mixing them with hexane (0.1520 g). Next, a portion (0.1 mL) of the formed solution was diluted with hexane (dilution factor of 1:10). A portion (exactly 2.5 µL) of the precursor solution was then directly drop cast onto CP (0.25 cm$^2$). After drying in air for 5 min, the electrode was annealed in an oven at 100 °C for 1 h. Finally, copper wire was connected to the anode by encapsulating the wire-exposed area with J-B Weld epoxy-steel resin, to afford an area of 0.25 cm$^2$ (Supplementary Fig. 24d).

**Electrochemical measurements**. Electrochemical measurements were carried out in a standard three-electrode system with the cathode and anode separated by a Nafion 117 membrane (DuPont). A Nafion proton exchange membrane was selected to concentrate the HO$_2^-$ anions, while Pt mesh (1 × 1 cm$^2$), Hg/HgO, and carbon sheets were used as the counter, reference, and working electrodes, respectively. All potentials in this study were measured against the Hg/HgO electrode and converted to the RHE reference scale by the equation $E$(V vs. RHE) = $E$(V vs. Hg/HgO) + 0.0592 × pH + 0.118. Prior to testing, the Nafion 117 membrane was wetted with distilled water for ~2 h to allow for sufficient swelling. Prior to electrolysis, the electrolyte solution was purged with high-purity O$_2$ (99.995%) for at least 10 min to ensure the removal of any residual air in the reaction system. During electrolysis, high-purity O$_2$ (99.995%) was continuously fed (at a rate of 50 cc min$^{-1}$) into the cathodic compartment with magnetic stirring. For comparison, electrolysis tests were also conducted in high-purity Ar (99.999%) saturated electrolyte solutions under the same experimental conditions. All current densities were normalised to the geometrical area of the electrodes.

**PEC measurements**. Photoelectrochemical measurements to confirm the activity (especially JV curves) were carried out in a standard three-electrode system using a half-cell reactor. Pt mesh (1 × 1 cm$^2$), Hg/HgO (1 M NaOH), and an O-BP/FM/

PSK photocathode were used as the counter, reference, and working electrodes, respectively. All potentials in this study were measured against the Hg/HgO reference electrode and converted to the RHE reference scale by the equation $E$(V vs. RHE) = $E$(V vs. Hg/HgO) + 0.0592 × pH + 0.118. PEC measurements to confirm the performance (especially to visualise the long-term stability) were carried out in a standard two-electrode system in a two-compartment reactor. The two-compartment reactor containing a 0.1 M KOH solution (7.5 mL) in both the anode and cathode chambers was separated by a Nafion 117 membrane, which was activated by maintaining it in deionised water for 2 h before the reaction. An a-NiFeO$_x$/CP anode and O-BP/FM/PSK photocathode were used as the counter and working electrodes, respectively.

Prior to photoelectrolysis, the electrolyte solution was purged with high-purity O$_2$ (99.995%) for at least 30 min (flow rate of 50 cc min$^{-1}$) to ensure the removal of any residual air in the reaction system. In addition, during photoelectrolysis, high-purity O$_2$ was continuously fed near the electrocatalyst surface into the cathodic compartment, with magnetic stirring, to overcome mass transfer limitation. For comparison, electrolysis tests were also conducted in high-purity (99.999%) Ar-saturated electrolyte solutions under the same experimental conditions. All current densities were normalised to the geometrical area of the photoelectrodes, and the scan rate during measurement was maintained at 5 mV s$^{-1}$. A 300 W xenon lamp was used to produce the simulated 1-sun irradiation with a water filter. The light intensity was adjusted at the FTO side (back-side illumination) to 1-sun by placing a silicon reference cell (certified by Newport Corporation; PEC-SI01, Peccell Technologies) parallel to the beam light of the xenon lamp.

**PEC H$_2$O$_2$ detection and its quantification**. The H$_2$O$_2$ concentration was determined after collecting aliquots of a desired volume at regular intervals from the catholyte solution and storing them in a refrigerator prior to measurement. The commonly employed N,N-diethyl-1,4-phenylene-diamine sulphate (DPD)–peroxidase (POD) method was used to determine the concentration at 551 nm. Briefly, DPD (50 mg, ≥98.0%, Sigma–Aldrich) and POD (5 mg, horse-radish, Sigma–Aldrich) were dissolved in 0.1 N H$_2$SO$_4$ (5 mL) and deionised water (5 mL), respectively, and stored in the dark at 5 °C. Finally, a 0.1 M sodium phosphate buffer (2.7 mL, ~pH 6.0), portions of the DPD (0.05 mL) and POD (0.05 mL) solutions, and the catholyte (0.2 mL) were mixed. Depending on the concentration of produced H$_2$O$_2$, the samples were diluted with 0.1 M KOH to avoid exceeding the detection limit. For its quantification, the standard calibration curve was plotted against the concentration and absorbance (Supplementary Fig. 29a, b) and, accordingly, the values of concentrated H$_2$O$_2$ produced were determined as a function of time. IPCE measurements were carried out using a 300 W xenon lamp as the light source and a monochromator with a bandwidth limit of 20 nm. The intensity of the reference light was calibrated prior to the IPCE measurements.

**Material characterisation**. A UV-vis spectrophotometer (UV-2600, Shimadzu) was used to determine the optical properties of the PSK thin films. SEM was performed using an SU-8220 scanning electron microscope (Hitachi), which was also used to determine the thickness of the all-solution processed layer. A JEOL JEM 2100F microscope was employed to record high-resolution TEM micrographs, and a JEM 2100 transmission electron microscope was used to record the TEM micrographs. Raman spectroscopy images were captured by a confocal Raman alpha300R instrument, while the surface areas of BP and O-BP were characterised using an ASAP 2420 physisorption analyser. A Zetasizer Nano ZS instrument (Malvern instruments) was used to determine the zeta-potentials, whereas the functional groups attached to O-BP were determined by XPS (k-alpha). The contact angles of the O-BP, BP, O-CNTs, and CNTs were measured by Phoenix 300 equipment. Finally, ultraviolet photoelectron spectroscopy (UPS) was per-formed on an ESCALAB 250XI system (Thermo Fisher) to elucidate the energy level positions.

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

## Acknowledgements

This research was supported by the National Research Foundation (NRF; 2021M3H4A1A0305138311, J.-W.J., 2019M1A2A2065612, J.-W.J., 2021R1C1C1012258, J.-W.J., 2017M1A2A2087630, J.-W.J., and 2017R1D1A1B03035450), and Research Fund (1.210033.01, J.-W.J.) of the UNIST (Ulsan National Institute of Science and Technology). This work is also supported by the Science Fellowship of the POSCO TJ Park Foundation, J.-W.J.

## Author contributions

R.M., D.O., and J.-W.J. proposed and conceived the research. R.M. prepared and characterised the perovskite and measured the photocathode performances. D.O. prepared and characterised the carbon materials and measured their performances. R.M., D.O., and J.-W.J. co-wrote the manuscript. J.-W.J. directed the research. All authors read and commented on the manuscript.

## Competing interests

The authors declare no competing interests.
