## [Peer review file · Nature Communications]

REVIEWER COMMENTS

Reviewer #1 (Remarks to the Author):

This manuscript reported the unassisted solar H₂O₂ production by combining an oxidised buckypaper as the H₂O₂ electrocatalyst with inorganic-organic hybrid (perovskite) photocathode, without the need for additional bias or sacrificial agents. This integrated photoelectrode system shows 100 % selectivity toward H₂O₂ and a solar-to-chemical conversion efficiency of ~1.204 %. When connecting with the a-NiFeOx/CP anode via a copper wire, continuous and stable H₂O₂ production of 407 ± 8.5 μmol cm⁻² was achieved for 12 h. Overall, the manuscript is well organized and written. Based on the novelty and significance of this work, acceptance of this work for Nature Communications is therefore recommended after addressing the following concerns.

- What is Field's metal? How it works as a protection layer for the PSK film from water penetration?

Please provide detailed descriptions.

- Raman spectra demonstrated that acidic treatment of CNT causes structural disorder, producing many defects on O-CNT. Nevertheless, it is actually difficult to observe such defects from TEM images (Supplementary Fig. 5).

- Zeta-potential results from Supplementary Fig. 7a demonstrated that O-CNT was more dispersible in N, N dimethylformamide (DMF), but why indicates that O-BP exhibits a more intimate interaction with water than BP? Besides, the authors should also provide the reason why a more negative zeta-potential in 0.1 M KOH indicates better H₂O₂ (HO₂⁻ in alkaline solution) generation with more facile product detachment.

- Some more recent important work about H₂O₂ production should be also cited in the Introduction part, such as Chem. Soc. Rev., 2020, 49, 6605-6631; Chem. Eng. J., 2021, 418, 129346.

Reviewer #2 (Remarks to the Author):

This manuscript mainly introduced the design of an inverted perovskite photocathode integrated with O₂ reduction electrocatalysts for hydrogen peroxide production. Also, the perovskite photocathode was connected with an OER NiFeOx anode in a two-electrode set-up to produce hydrogen peroxide with no bias. The performance of using perovskite photoelectrode with O-BP catalysts exceeded the other reported results and was proven to be stable for 20 hours. Overall, the paper is generally well organized and does a good job to present the idea of combining perovskite solar cells and ORR electrocatalysis.

However, I have several concerns:

1. Lack of Significant Novelty. The use of field metal in protecting halide perovskites has been demonstrated in literature. The main novelty is to coat a layer of buckypaper (BP) on top of the protected halide perovskite for H₂O₂ production. This is a quite incremental progress of engineering

2. From Figure 2d, why does the BP without oxidation process has an oxygen peak from XPS? Compared with Supplementary Figure 6a, CNTs which is the precursor of BP do not have any oxygen peak.

3. On page 7, line 137, "the selectivities of both BP and O-BP decreased because the produced H₂O₂ could be further reduced to water at these potentials" Considering the reduction potential of H₂O₂ to H₂O (+1.77 V vs. RHE), the applied bias can always further reduce H₂O₂ to H₂O, so it cannot explain the selectivity decrease issue. Increase reduction of H₂O₂ because of the faster accumulation of H₂O₂ in electrolyte at more negative bias can better explain this.

Reviewer #3 (Remarks to the Author):

This manuscript reported an oxidised buckypaper-integrated perovskite photocathode for solar H₂O₂ production. Using this integrated device, the authors achieved a high solar-to-chemical conversion efficiency of 1.22%. The result was quite interesting and may promote the development of device for solar H₂O₂ production. The following queries should be well solved.

1. Fig. 1 and Fig. 3. The integrated perovskite photocathode was composed by 5 layer of materials with different thickness. Does the thickness of each layer have impact of the catalysis performance and how to determine the optimized thickness for each layer?
2. Fig. 1. How does the electron, hole, and mass (e.g., proton) transport in the whole system? A clear pathway should be provided to help the reader better understand the catalysis process.
3. What are the electrolytes for photocathode and anode respectively? 0.1 M KOH for both? The authors did not clearly mention the relevant important information in the Experiment section. I thought the experiments were conducted in alkaline media, where the H₂O₂ is in the form of peroxide ion. How about the performance in neutral or acid media?
4. Following the above question, unlike the electrolyte-free H₂O₂ production system, how can the produced H₂O₂ be separated from the electrolyte for practical use?
5. The author claimed the perovskite photocathode was very stable, by analysing the current within a given test period. More characterizations (e.g., morphology and chemical states) should be conducted to further support this point.
6. Line 250, what is the SCC after 1h reaction? It seems the FE% slightly declined after 6 hour, so the SCC% might also dropped after long time test.
7. Supplementary Table. The electrolyte for all systems should be provide to better compare the performance. Also there is a Typo. The year number in the last row was not correct.

Point-by-point replies to the reviewer comments

Response to Comments of Reviewer #1

General Comment: This manuscript reported the unassisted solar H₂O₂ production by combining an oxidized buckypaper as the H₂O₂ electrocatalyst with inorganic-organic hybrid (perovskite) photocathode, without the need for additional bias or sacrificial agents. This integrated photoelectrode system shows 100 % selectivity toward H₂O₂ and solar-to-chemical conversion efficiency of ~1.204 %. When connecting with the a-NiFeO_x/CP anode via a copper wire, continuous and stable H₂O₂ production of $407 \pm 8.5 \mu\text{mol cm}^{-2}$ was achieved for 12 h. Overall, the manuscript is well organized and written. Based on the novelty and significance of this work, acceptance of this work for Nature Communications is therefore recommended after addressing the following concerns.

Response: We appreciate the reviewer's positive and constructive comments. We have tried to address the reviewer's concerns by performing additional relevant experiments. The reviewer's valuable and detailed comments have helped us improve the quality of our manuscript. Our responses to the reviewer's comment have been presented below.

Comment 1: What is Field's metal? How it works as a protection layer for the PSK film from water penetration? Please provide detailed descriptions.

Response: We thank the reviewer for the significant and detailed comments. Briefly, the Field's metal (FM) is a eutectic alloy composed of indium, bismuth, and tin with the compositions of 51 %, 32.5 %, and 16.5 %, respectively. Owing to the eutectic nature, the alloy melts at a lower temperature (~ 62 °C) than those of its individual constituents (indium: ~ 156.6 °C; bismuth: ~ 271.4 °C; tin: ~ 231.9 °C) and solidifies at room temperature. Using this property, we first attached FM to a PSK layer at 65 °C (which is higher than the melting point of FM but less than the degradation temperature of perovskite, [*Chem. Rev.* 119, 5, 3418, (2019)]), which then formed a close and tight junction between the PSK film and FM during the solidifying process of FM at the ambient temperature (~ 27 °C). This solidified FM has a closely packed atomic structure like other metals, thus 1 mm of FM layer was enough to inhibit the electrolyte intrusion. In this way, the PSK films become impenetrable to any water-based electrolytes, which otherwise can destroy the films.

Comment 2: Raman spectra demonstrated that acidic treatment of CNT causes structural disorder, producing many defects on O-CNT. Nevertheless, it is actually difficult to observe such defects from TEM images (Supplementary Fig. 5).

Response: We thank the reviewer for the critical and detailed comments. We definitely agree with the reviewer's opinion that TEM images in the previous manuscript are not sufficient to show the defect structures of O-CNTs. In the Revised Supplementary Fig. 5, we carefully performed the high resolution TEM (HR-TEM) analysis again to more clearly show the structural differences between the CNTs and O-CNTs.

Revised Supplementary Fig. 5 | High-resolution transmission electron micrographs. a, CNT and b, O-CNT. a,b, Scale bar, 10 nm.

Comment 3: Zeta-potential results from Supplementary Fig. 7a demonstrated that O-CNT was more dispersible in N, N dimethylformamide (DMF), but why indicates that O-BP exhibits a more intimate interaction with water than BP? Besides, the authors should also provide the reason why a more negative zeta-potential in 0.1 M KOH indicates better H₂O₂ (HO₂⁻ in alkaline solution) generation with more facile product detachment.

Response: We thank the reviewer for the significant and detailed comments. We compared the zeta-potential of the O-CNTs and CNTs in DMF (Revised Supplementary Fig. 8c), because it is well-known that more negative zeta potential of carbon materials in polar solvent such as DMF indicate that they possess a larger amount of functional groups, which results in a more intimate interaction with water [*Journal of Materiomics* **4**, 108, (2018); *Nature Nanotechnol.* **3**, 101, (2008)]. However, as the reviewer commented, to directly see the interaction between the O-CNTs (CNTs) and water, we additionally performed the zeta-potential measurements of the O-CNTs and CNTs in water as shown in Revised Supplementary Fig. 8a. The result shows that the O-CNTs have a more negative zeta-potential than the CNTs in water, which demonstrates successful surface modification with oxygen functional groups; this results in a more intimate interaction with water. This trend was the same in a basic solution (0.1 M KOH, Revised Supplementary Fig. 8b). To further prove that the O-CNTs are more hydrophilic than the CNTs, we performed contact angle measurements with water droplets on the surfaces of O-BP (buckypaper fabricated from O-CNTs)

and BP (buckypaper fabricated from CNTs) as shown in the Revised Supplementary Fig. 8d,e. The contact angles of O-BP and BP were 25° and 134°, respectively, indicating that the O-CNTs are more hydrophilic than the CNTs; this shows that the O-CNTs will have a more intimate interaction with water than the CNTs.

The reason why a more negative zeta-potential in 0.1 M KOH indicates better H₂O₂ generation with more facile product detachment is due to the repulsive interaction between the negative charge of the product (HO₂⁻) in an alkaline solution and the negative charge of the surface functional groups of the O-CNTs (CNTs) catalyst [*Energy Environmental Sci.* **13**, 238, (2020)]. Because O-CNTs have more surface functional groups and a more negative charge than the CNTs, as denoted above, the repulsive interactions between the product of HO₂⁻ and the surface of the O-CNTs electrocatalyst are higher than those between HO₂⁻ and the CNTs surface, which results in more facial product detachment.

Revised Supplementary Fig. 8 | Zeta-potentials of the carbon materials. Zeta-potentials of the CNTs and O-CNTs in **a**, deionised water, **b**, KOH, and **c**, DMF. A higher absolute zeta-potential value indicates better dispersivity. The more negatively charged O-BP woven by the O-CNTs facilitates the unhinging of the generated HO₂⁻ anions. **d**, **e**, Contact angle measurements of O-BP (**d**) and BP (**e**).

Added Contents:

Manuscript Text, Page number 8, line number 144

⇒ “... deionised water and *N,N*-dimethylformamide (DMF), indicating that O-BP should exhibit a more intimate interaction with water than BP (Supplementary Fig. 8a-c). This was also verified by contact angle measurements in which the contact angle of O-BP and BP were 25° and 134°, respectively, indicating that O-CNTs are more hydrophilic than CNTs (Supplementary Fig. 8d,e).

Manuscript Text, Page number 8, line number 151

⇒ due to the higher repulsive interactions between the negatively charged functional groups on the O-CNTs and the negatively charged HO₂⁻ species in aqueous 0.1 M KOH (CNTs have less functional groups than O-CNTs).

Manuscript Text, Page number 27, line number 506

⇒ The contact angles of the O-BP, BP, O-CNTs, and CNTs were measured by Phoenix 300 equipment.

Comment 4: Some more recent important work about H₂O₂ production should be also cited in the Introduction part, such as *Chem. Soc. Rev.*, 2020, 49, 6605-6631; *Chem. Eng. J.*, 2021, 418, 129346.

Response: We thank the reviewer for letting us know the important papers to be cited. We have added the [*Chem. Soc. Rev.*, 49, 6605, (2020); *Chem. Eng. J.* 418, 129346 (2021)] references in the revised manuscript.

Manuscript Text, Page number 3, line number 48, references 1–3

“... [*Nat. Commun.* 10, 5123, (2019); *Nature Commun.* 7, 11407 (2016); *Electrochimica Acta* 82, 493, (2012)].”

⇒ ... [*Chem. Soc. Rev.*, 49, 6605, (2020); *Nat. Commun.* 7, 11407 (2016); *Electrochimica Acta* 82, 493, (2012)].

Manuscript Text, Page number 3, line number 57, reference 9–11

~~“...[*Small* **16**, e1902845 (2020); *ACS Catal.* **10**, 7495-7511 (2020)].”~~

⇒ ...[*Small* **16**, e1902845 (2020); *ACS Catal.* **10**, 7495-7511 (2020); *Chem. Eng. J.* **418**, 129346 (2021)].

Response to Comments of Reviewer #2

General Comment: This manuscript mainly introduced the design of an inverted perovskite photocathode integrated with O₂ reduction electrocatalysts for hydrogen peroxide production. Also, the perovskite photocathode was connected with an OER NiFeOx anode in a two-electrode set-up to produce hydrogen peroxide with no bias. The performance of using perovskite photoelectrode with O-BP catalysts exceeded the other reported results and was proven to be stable for 20 hours. Overall, the paper is generally well organized and does a good job to present the idea of combining perovskite solar cells and ORR electrocatalysis.

Response: We thank the reviewer for the positive comments and encouragement. To fully address the reviewers' concerns, we have presented point-by-point responses after performing additional relevant experiments. We thank the reviewer for their critical and insightful comments that have helped us improve the quality of our manuscript.

Comment 1: Lack of Significant Novelty. The use of Field's metal in protecting halide perovskites has been demonstrated in the literature. The main novelty is to coat a layer of buckypaper (BP) on top of the protected halide perovskite for H₂O₂ production. This is quite incremental progress of engineering.

Response: We thank the reviewer for the sound criticism. We have tried to explain the novelty of our work in two aspects. First, this work is the first to apply perovskite (PSK) materials to produce solar H₂O₂ efficiently. In previous works, the solar-to-H₂O₂ conversion (SCC) efficiency was limited by the low performance of the inorganic semiconductor-based photoelectrodes due to their poor charge transfer characteristics and relatively large band gaps (band gap ~2.4 eV- 3.2 eV). In this work, by using a PSK (band gap of ~ 1.57 eV) for solar H₂O₂ production, we achieved record-high SCC efficiency (1.463 % after 2 h, revised Fig. 4b). Furthermore, it also needs to be noted that this PSK-based photoelectrode showed high stability (without sign of degradation for 45 h, Revised Supplementary Fig. 25), which is very unusual for PSKs applied for solar fuel production (they lose their activities within a few minutes or after only several hours [*Nat. Commun.* **7**, 12555, (2016); *ACS energy Lett.* **4**, 293, (2019)]). We also verified that the visual morphology, crystal structure, and electronic distribution of O-BP/FM/PSK was practically the same after stability tests (45 h) by scanning electron microscopic (SEM), X-ray diffraction (XRD), and X-ray photoelectron spectroscopy (XPS), in the revised manuscript.

The second novel aspect of our work is that we applied FM and O-BP to form a very intimate junction between the PSK photoelectrode and O-BP H₂O₂ electrocatalyst, as well as passivation layers to protect the PSK photoelectrode. To fabricate the high-performance and selective electrocatalyst-loaded photoelectrode, the intimate junction formation between the photoelectrode and electrocatalyst has been considered crucial. In other words, although individually the photoelectrode and electrocatalyst shows good performance, if there is not proper and intimate

junction between them, the performance and stability of the electrocatalyst-loaded photoelectrode could be very low due to a large charge transfer resistance occurring at the interfacial junction [Science 344, 6187, 1005, (2014); Chem. Soc. Rev. 48, 4979, (2019)]. As reviewer commented, several groups have already applied FM as a protection layer for PSK. However, in this work, we reported that FM can be applied as an efficient intermediate layer (as well as the protection layer), which makes an intimate and very tight junction formation between the PSK photoelectrode and O-BP H₂O₂ production electrocatalyst (Added Supplementary Fig. 19). In addition, not only did we apply O-BP as a H₂O₂ electrocatalyst for the first time, but we also revealed for the first time that electrocatalysts in the form of buckypaper (sheet-type) are favourable regarding the formation of an intimate junction with a PSK photocathode, as compared to a conventional powder-type electrocatalyst (Added Supplementary Fig. 21). Furthermore, we also verified that the O-BP electrocatalyst can act as another passivation layer to protect the PSK layer when it is combined with FM (Added Supplementary Fig. 20). Using this strategy, we succeeded in stabilizing a very unstable PSK layer with record high solar H₂O₂ production performance. We believe that this strategy can not only be applied to other solar H₂O₂ production works but can also be applied to produce other solar fuels, especially when unstable materials like PSKs or organics are used as the photoactive layer.

The detailed response to the reviewer's comment and the explanation of additional experiments during the revision are as follows:

In this work, FM not only acts as a rigid passivation layer preventing the permeation of water, but also acts as an intermediate layer that helps in the formation of the intimate junction, facilitating the facile charge transfer between the PSK photocathode and O-BP H₂O₂ electrocatalyst. Using the properties of FM (i.e., the fact that it melts above its melting point of 62 °C where degradation of the perovskite layer does not occur [Chem. Rev. 119, 5, 3418, (2019)] and is solidified at room temperature), we attached the PSK photocathode with the O-BP H₂O₂ electrocatalyst at 65 °C and solidified it at ~ 27 °C in a glove box, thus making a very close and tight junction between them at the atomic level. To figure out the effect of the FM layer on the performance of the O-BP-loaded PSK, we compared its performance with and without FM. Without using the FM as a junction layer, it shows a negligible photocurrent (~ 110 μA cm⁻² at 0 vs. RHE) and poor stability, as shown in the Added Supplementary Fig. 19.

It also needs to be noted that applying the sheet-type O-BP electrocatalyst (instead of using a conventional powder-type O-CNTs electrocatalyst) is crucial in making the compact and intimate junction with the PSK photocathode through FM. To experimentally show the importance of utilizing O-BP in the conformal junction formation, we compared the performance of an O-BP-loaded FM cathode with that of an O-CNTs powder-loaded one. As shown in the Added Supplementary Fig. 21a, the O-CNTs powder on the FM induced a negative shift in its onset potential by 170 mV compared to the O-BP on FM. This onset potential change indicated poor adhesion between the O-CNTs and FM [Chem. Rev. 110, 5790, (2010); Adv. Funct. Mater. 24, 4763, (2014)]. As shown in the Added Supplementary Fig. 21b, the initial current density of O-

CNTs/FM at 0.6 V vs. RHE in 0.1 M KOH (O_2 purging) was only -0.5 mA cm^{-2} , which was approximately four times less than that of O-BP/FM. Furthermore, it lost its activities within a few minutes due to the detachment of the O-CNT powder from the FM. Contrastingly, O-BP/FM showed stable performance. These results indicate that O-BP is more suitable to form an intimate and stable junction with a PSK photoelectrode via FM. To further demonstrate the importance of O-BP in stabilizing the PSK layer, we conducted additional stability and the performance tests without loading O-BP (FM/PSK) (Added Supplementary Fig. 20). As a result, the stability and performance of FM/PSK were poor, indicating that O-BP not only serves as a selective H_2O_2 production electrocatalyst, but also acts as important passivation layer when combined with FM. Additionally, we also optimized the thickness of O-BP with ETL, perovskite, and HTL to further optimize the performance and stability according to reviewer 3's comments (Added Supplementary Fig. 11, Added Supplementary Fig. 12, and Added Supplementary Fig. 17). After these optimizations, we achieved higher photocurrent density and SCC efficiency of 1.463 % after 2 h (revised Fig. 3 and Fig. 4a,b; before the revision, SCC efficiency was 1.204 %) with better stability of 45 h (Revised Supplementary Fig. 25; there was no sign of degradation during the stability tests) (before the revision, we tested the stability for 20 h). We further verified that the visual morphology, crystal structure, and electronic distribution of O-BP/FM/PSK were practically the same by performing SEM, XRD, and XPS analyses, respectively (as seen in Added Supplementary Fig. 26).

Added Supplementary Fig. 19 | Performance measurement of the O-BP/PSK photocathode. **a**, LSV measurements of the O-BP/PSK device without the FM as an intimate junction layer, demonstrating very low photocurrent density over the entire potential region. **b**, Stability measurements at 0 V vs. RHE show degradation within 30 min. Experiments conducted in a 0.1 M KOH solution (pH ~13.17) under ambient conditions at a scan rate of 5 mV s^{-1} and with a continuous O_2 supply.

Added Supplementary Fig. 21 | Electrochemical performance measurement of powder and sheet type carbon catalysts. a, b, ORR activity (a) and chronoamperometry measurements (b) of sheet-type (O-BP) and powder-type (O-CNTs) carbon materials combining with FM. The weight of the powder-type carbon is the same as that of the sheet-type carbon (~ 1.3 mg in 0.25 cm^2).

Added Supplementary Fig. 20 | Performance measurement of the FM/PSK photocathode. a, LSV measurements of the FM/PSK device without the integration of the O-BP as an electrocatalyst. b, Stability measurements at 0.6 V vs. RHE show an unstable photocurrent during the reaction. a, b, Experiments conducted in a 0.1 M KOH solution (pH ~ 13.17) under ambient conditions at a scan rate of 5 mV s^{-1} and with a continuous O_2 supply.

Added Supplementary Fig. 11 | Cross-sectional scanning electron microscopy (SEM) images of O-BP. a, c, e, Cross-sectional SEM images of O-BP in amounts of 50 mg (a), 100 mg (c), and 150 mg (e). b, d, f, Corresponding cyclic voltammograms of O-BP in amounts of 50 mg (b), 100 mg (d), and 150 mg (f) for determining capacitance in addition to the oxygen reduction reaction (ORR) activity.

Added Supplementary Fig. 12 | Optimisation for O-BP thickness on ORR activity. a,b, Optimisation for ORR activity (a) and the results of 12 h of stability tests (b) for 50 mg (dark grey), 100 mg (red), and 150 mg (blue) of O-BP. Every experiment was conducted at + 0.6 V (vs. RHE) under an O_2 environment.

Added Supplementary Fig. 17 | Cross-sectional scanning electron microscopic images of MAPbI₃. **a–c**, Variation in the thickness of the HTL (with other layer conditions kept constant) due to a change in the spin-casting speed (rpm). Scale bars, 1 μm. **d**, The corresponding effect of the thickness on the performance of the integrated O-BP/Field’s metal (FM)/PSK device. **e–g**, Variation in the thickness of photoabsorber (with other layer conditions kept constant) due to a change in the spin-casting speed (rpm); this also reduced the substrate temperature and thus changed the morphology. Scale bars, 1 μm. **h**, Corresponding effect of thickness on the performance, which shows that optimised thickness is necessary when the precursor ink is dispensed. **i–k**, Variation in the thickness of the electron transfer layer (ETL) (with other layer conditions kept constant) due to a change in the spin-casting speed (rpm). Scale bars, 1 μm. **l**, Since the ETL was not very thick, the variation in the performance was negligible as compared to

the changes observed with the variations caused by changing the thickness of the HTL and PSK photoabsorber.

Added Supplementary Table 1 | Optimisation process conditions of all-solution processed layers including the active catalyst

Spin coating parameters		Thickness	Jsc (mA cm ⁻²)	Comments/Onset
Ramp	time			
HTL thickness optimisation				
1500	60 s	94 nm	~ -5 mA cm ⁻²	Too thick HTL
4000	60 s	40 nm	~ -11.15 mA cm ⁻²	Optimised
6500	60 s	17 nm	~ -8 mA cm ⁻²	Non-covered area
Photo-absorber thickness optimisation				
1000	20 s	600 nm	~ -8.8 mA cm ⁻²	1.64 V vs. RHE
2200	20 s	365 nm	~ -11.15 mA cm ⁻²	1.77 V vs. RHE
4000	20 s	200 nm	~ -3 mA cm ⁻²	1.75 V vs. RHE
ETL thickness optimisation				
1000	30 s	55 nm	~ -7.8 mA cm ⁻²	No difference in the onset value
2000	30 s	40 nm	~ -11.15 mA cm ⁻²	
3000	30 s	34 nm	~ -9.2 mA cm ⁻²	
O-BP thickness optimisation at 0.6 V vs. RHE determining the mass activity (A g⁻¹)				
Loading amount		Thickness		
50 mg		~ 92 μm	1.924 mA cm ⁻²	0.67 A g ⁻¹
100 mg		~ 206 μm	1.964 mA cm ⁻²	0.34 A g ⁻¹
150 mg		~ 302 μm	2.005 mA cm ⁻²	0.23 A g ⁻¹

Revised Fig. 3 | Characterisation and ORR activity of the O-BP-integrated PSK photocathode. **a,b**, Top-view and cross-sectional scanning electron micrographs of the formed MAPbI₃ PSK showing the thickness of the individual layers. **c**, J - E responses of the O-BP electrocatalyst, Field's metal (FM)/PSK photocathode, and integrated O-BP/FM/PSK photocathode device, exhibiting an anodic shift. Integrated O-BP/FM/PSK photocathode devices were simulated under 1-sun and an air mass of 1.5 G, with the averaged cyclic voltammograms (Avg. CV) scans under light and no irradiation (red arrow indicates the scan direction during the experiment) in the presence of an O₂ supply. **d**, Wavelength-dependent spectral response (i.e., incident-photon-to-current efficiency (IPCE)) and the corresponding integrated photocurrent density of the integrated device at 0 V vs. RHE. **e**, J - E response of the *a*-NiFeO_x/CP anode without stirring. Inset depicts the exact overpotential for the OER. **f**, Chronopotentiometry test of the *a*-NiFeO_x/CP anode conducted in a three-electrode set-up. **c-f** Experiments conducted in a 0.1 M KOH solution (pH ~13.17) under ambient conditions with a scan rate of 5 mV s⁻¹. All scale bars equal 1 μm.

Revised Fig. 4 | Unassisted solar H₂O₂ production with integrated photocathode system. | **a**, Operating point, obtained from the overlap of the $a\text{-NiFeO}_x/\text{CP}$ anode and the integrated system of the O-BP/FM/PSK photocathode, for the generated solar H₂O₂ affords a photocurrent of ~ 2.51 mA cm⁻² for generating solar H₂O₂ in the catholyte. **b**, In a two-electrode set-up under simulated 1-sun conditions, the unassisted H₂O₂ production (n) of our integrated O-BP/FM/PSK device could continuously produce $\sim 442 \pm 15.2$ $\mu\text{mol cm}^{-2}$ of H₂O₂ for 12 h, without any degradation, at 0 V. **c**, Graph showing the generated rate of solar H₂O₂ on the left y-axis, while the corresponding solar-to-chemical conversion efficiency (SCC; %) has been marked on the right y-axis (Supplementary Table 2). The highest SCC efficiency peaked after 2 h of reaction, also demonstrated in Supplementary Fig. 28.

Added Supplementary Fig. 25 | Unassisted solar H₂O₂ generation at 0 V vs. counter electrode. Long-term stability tests of our integrated O-BP/FM/PSK photocathode under short-circuit conditions. Experiments were conducted in a 0.1 M KOH solution (pH ~13.17) under ambient conditions with a continuous O₂ supply.

Added Supplementary Fig. 26 | Characterisation of the integrated O-BP/FM/PSK photocathode after measuring the long-term stability test. **a**, XRD patterns of the carbon materials before and after stability tests. **b,c**, SEM images of the O-BP before (**b**) and after (**c**) the long-term stability tests showing no accumulation of the incoming electrons at the interface rather than the rapid consumption towards the conversion into H_2O_2 . **d, e**, XPS surveys of the carbon materials before and after stability tests. All scale bars equal $1 \mu\text{m}$.

Modified: Manuscript Text, Page number 17, line number 314

“Moreover, the performance of the O-BP/FM/PSK photocathode did not decline during 20h of testing, with fluctuations in the obtained photocurrent density after 12 h, mainly due to the addition of the electrolyte (Supplementary Fig. 17).”

⇒ Moreover, the performance of the O-BP/FM/PSK photocathode did not show any sign of decrease during the 45 h of stability tests, as shown in Supplementary Fig. 25. XRD, scanning electron microscopy (SEM), and XPS analyses of the O-BP/FM/PSK photocathode after the stability tests (Supplementary Fig. 26) showed no significant changes in its crystal structure, morphology, or electronic distribution.

Added contents:

Manuscript Text, Page number 4, line number 74

⇒ It also needs to be noted that here we applied PSK-based photoelectrode as well as O-BP electrocatalyst for solar H₂O₂ production for the first time.

Manuscript Text, Page number 9, line number 172

⇒ The thickness of O-BP did not affect its performance or stability if the value was greater than ~100 μm, as shown in Supplementary Fig. 11 and Supplementary Fig. 12.

Manuscript Text, Page number 12, line number 214

⇒ We optimised the thicknesses of the individual layers, including the HTL, PSK photo-absorber, and ETL, to enhance the performance of the O-BP/FM/PSK photocathode, as shown in Supplementary Fig. 17 and Supplementary Table 1.

Manuscript Text, Page number 13, line number 233

⇒ The O-BP/PSK photocathode without use of FM between O-BP and PSK layers exhibits a very low photocurrent density of 110 μA cm⁻² at 0 V vs. RHE and lost its photoactivity within 30 min (Supplementary Fig. 19). In addition, less than 12 % of the photocurrent of the FM/PSK photoelectrode remained after 10 h at 0.6 V vs. RHE (Supplementary Fig. 20). These results show the importance of FM and O-BP in forming an intimate junction between the PSK layer and O-BP to stabilise the PSK layer. We then compared the performances of the O-BP (sheet)/FM electrocatalyst and an O-CNTs (powder)/FM electrocatalyst to investigate if a sheet or a powder would more suitably combine with FM to give an electrocatalyst with better performance and stability (Supplementary Fig. 21). There was a negative shift in the onset potential of O-CNTs/FM of 170 mV relative to that of O-BP/FM (Supplementary Fig. 21a). The initial current density of O-CNTs/FM at 0.6 V vs. RHE in 0.1 M KOH (O₂ purging) was only -0.5 mA cm⁻², which was approximately four times less than that of O-BP/FM. Furthermore, the O-CNTs/FM electrocatalyst lost its activity within a few minutes due to the

detachment of the O-CNTs powder from the FM; contrastingly, O-BP/FM exhibited stable performance (Supplementary Fig. 21b). These results indicate that a sheet forms a more intimate and stable junction with FM than a powder, making the former more suitable for use in a H₂O₂ electrocatalyst.

Comment 2: From Figure 2d, why does the BP without oxidation process has an oxygen peak from XPS? Compared with Supplementary Figure 6a, CNTs which are the precursor of BP do not have any oxygen peak.

Response: We thank the reviewer for the critical and detailed comments. It has been known that several gases and oxygen-containing species can be easily absorbed on BP during the filtration of process of CNTs [*Chem Rev* **119**, 599, (2019); *Angew Chem Int Ed Engl* **47**, 6550-6570, (2008)] [*Science*, 287, 1801, (2000); *Carbon*, **44**, 2155, (2006); *Mater Lett*, **122**, 281, (2014); *J Am Chem Soc* **125**, 11329, (2003)]. For this reason, many other research groups also reported that an oxygen peak was observed when the XPS analysis of BP was conducted [*Nat Mater*, **19**, 189, (2020); *Angew Chem Int Ed Eng* **55**, 3952, (2016)]. To verify this further, we conducted thermogravimetric analysis (TGA) and contact angle measurements. Added Supplementary Fig. 7 clearly shows that for the non-functionalised carbon materials (BP and CNTs) there was almost no weight loss up to 500 °C, where the detachment of functional groups takes place; meanwhile, for the functionalised (O-BP and O-CNTs) carbon materials, there was a weight loss of about 10 % due to the detachment of functional groups. These results indicate that the oxygen peak of BP is not due to its functional groups, but rather adsorbed oxygen-containing impurities. It also needs to be noted that the amount of oxygen-containing impurities absorbed by BP is very negligible compared with the amount of functional groups of O-BP, as shown in the contact angle measurements of both BP and O-BP with TGA results above (revised Supplementary Fig. 8d,e). The contact angles of water droplets on O-BP and BP were 25° and 134°, respectively, indicating that BP is very hydrophobic because it possesses a small amount of absorbed oxygen-containing chemicals compared with O-BP.

Added Supplementary Fig. 7 | Thermogravimetric analyses. a, CNTs with BP. b, O-CNTs with O-BP.

Revised Supplementary Fig. 8 | Zeta-potentials of the carbon materials. Zeta-potentials of the CNTs and O-CNTs in **a**, deionised water, **b**, KOH, and **c**, DMF. A higher absolute zeta-potential value indicates better dispersivity. The more negatively charged O-BP woven by the O-CNTs facilitates the unbinding of the generated HO_2^- anions. **d**, **e**, Contact angle measurements of O-BP (**d**) and BP (**e**).

Added Contents:

Manuscript Text, Page number 7, line number 126

⇒ The oxygen peaks that appear for BP are due to surface contamination with oxygen-containing species, which are likely adsorbed during the filtration process of the CNT-containing solution [J. Am. Chem. Soc. **125**, 11329, (2003); Science, 287, 1801, (2000); Carbon, **44**, 2155, (2006); Mater Lett, **122**, 281, (2014)]. Thermogravimetric (TGA) analysis (Supplementary Fig. 7) showed that there was no weight loss in the non-functionalised carbon materials, including the BP and CNTs, while a weight loss of about 10 % was observed for the functionalised (O-BP and O-CNTs) carbon materials due to detachment of functional group; this weight loss occurred when the temperature was increased up to 500 °C. This result indicates that the oxygen peak observed for BP originates from oxygen-containing impurities rather than functional groups, and its absolute amount is very low. Contact angle measurements of BP and O-BP also support this (Supplementary Fig. 8).

Manuscript Text, Page number 8, line number 144

⇒ "... deionised water and *N,N*-dimethylformamide (DMF), indicating that O-BP should exhibit a more intimate interaction with water than BP (Supplementary Fig. 8a-c). This was also verified by contact angle measurements in which the contact angle of O-BP and BP were 25° and 134°, respectively, indicating that O-CNTs are more hydrophilic than CNTs (Supplementary Fig. 8d,e).

Manuscript Text, Page number 27, line number 506

⇒ The contact angles of the O-BP, BP, O-CNTs, and CNTs were measured by Phoenix 300 equipment.

Comment 3: On page 7, line 137, “the selectivities of both BP and O-BP decreased because the produced H_2O_2 could be further reduced to water at these potentials” Considering the reduction potential of H_2O_2 to H_2O (+1.77 V vs. RHE), the applied bias can always further reduce H_2O_2 to H_2O , so it cannot explain the selectivity decrease issue. Increase reduction of H_2O_2 because of the faster accumulation of H_2O_2 in the electrolyte at a more negative bias can better explain this.

Response: We very thank the reviewer for pointing out our improper explanation. We fully agree with reviewer’s comments. We have revised the manuscript according to reviewer’s comments.

Modified: Manuscript Text, Page number 9, line number 165

“...~~the produced H_2O_2 could be further reduced to water at these potentials~~ the degradation rate of H_2O_2 increased due to the faster accumulation rate of H_2O_2 at these high negative potentials (Supplementary Fig. ~~8e,d~~ 9c,d and Supplementary Fig. 910).

Response to Comments of Reviewer #3

General Comment: This manuscript reported an oxidised buckypaper-integrated perovskite photocathode for solar H₂O₂ production. Using this integrated device, the authors achieved a high solar-to-chemical conversion efficiency of 1.22%. The result was quite interesting and may promote the development of a device for solar H₂O₂ production. The following queries should be well solved.

Response: We appreciate the reviewer's encouraging and insightful comments. We have presented point-by-point responses to the comments. We have also conducted additional experiments to address them. The insightful comments have helped us improve the quality of our manuscript.

Comment 1: Fig. 1 and Fig. 3. The integrated perovskite photocathode was composed by 5 layer of materials with different thickness. Does the thickness of each layer have impact of the catalysis performance and how to determine the optimised thickness for each layer?

Response: We thank the reviewer for the fruitful and significant comments. The thickness of each layer is a significant factor in determining the performance of the overall integrated system, as reviewer commented. Thus, we optimized the thickness of each layer, including the HTL, PSK layer, ETL and oxidized buckypaper (O-BP). However, the thickness of FM was not easy to control because it presents as liquid at the temperature used to attach the BP electrocatalyst to the PSK photocathode (Added Supplementary Fig. 11, 12, 17, Added Supplementary Table 1). Thus, to optimize the thickness of layer, we controlled the spin-casting speed rather than changing the concentrations of the solutions. We measured the thickness of each layer by taking cross-sectional scanning electron microscopic images. Upon these optimizations, the photocurrent density of the optimized oxidised buckypaper-integrated perovskite (denoted as O-BP/FM/PSK) photocathode was increased from $-10.18 \text{ mA cm}^{-2}$ to $-11.15 \text{ mA cm}^{-2}$ at 0 V vs. RHE, as shown in revised Fig. 3c. Furthermore, the solar to H₂O₂ conversion (SCC) efficiency was also increased from 1.204% to 1.463 %, when this O-BP/FM/PSK photocathode was combined with NiFeO_x anode (revised Fig. 4).

Layer 1- PEDOT:PSS (HTL)

Layer 2- MAPbI₃ (Photo-absorber) fabricated by two-step spin coating; PbI₂.

Layer 3- PCBM (ETL)

Layer 4- Active catalyst (Oxidised- buckypaper, O-BP)

Added Supplementary Fig. 17 | Cross-sectional scanning electron microscopic images of MAPbI₃. **a–c**, Variation in the thickness of the HTL (with other layer conditions kept constant) due to a change in the spin-casting speed (rpm). Scale bars, 1 μm . **d**, The corresponding effect of the thickness on the performance of the integrated O-BP/Field’s metal (FM)/PSK device. **e–g**, Variation in the thickness of photoabsorber (with other layer conditions kept constant) due to a change in the spin-casting speed (rpm); this also reduced the substrate temperature and thus changed the morphology. Scale bars, 1 μm . **h**, Corresponding effect of thickness on the performance, which shows that optimised thickness is necessary when the precursor ink is dispensed. **i–k**, Variation in the thickness of the electron transfer layer (ETL) (with other layer conditions kept constant) due to a change in the spin-casting speed (rpm). Scale bars, 1 μm . **l**, Since the ETL was not very thick, the variation in the performance was negligible as compared to

the changes observed with the variations caused by changing the thickness of the HTL and PSK photoabsorber.

Added Supplementary Fig. 11 | Cross-sectional scanning electron microscopy (SEM) images of O-BP. a, c, e, Cross-sectional SEM images of O-BP in amounts of 50 mg (a), 100 mg (c), and 150 mg (e). **b, d, f,** Corresponding cyclic voltammograms of O-BP in amounts of 50 mg (b), 100 mg (d), and 150 mg (f) for determining capacitance in addition to the oxygen reduction reaction (ORR) activity.

Added Supplementary Fig. 12 | Effect of varied O-BP thickness on ORR activity. a,b, Optimisation for ORR activity (a) and the results of 12 h of stability tests (b) for 50 mg (dark grey), 100 mg (red), and 150 mg (blue) of O-BP. Every experiment was conducted at + 0.6 V (vs. RHE) under an O₂ environment.

Added Supplementary Table 1 | Optimisation process conditions of all-solution processed layers including the active catalyst

Spin coating parameters		Thickness	J _{sc} (mA cm ⁻²)	Comments/Onset
Ramp	time			
HTL thickness optimisation				
1500	60 s	94 nm	~ -5 mA cm ⁻²	Too thick HTL
4000	60 s	40 nm	~ -11.15 mA cm ⁻²	Optimised
6500	60 s	17 nm	~ -8 mA cm ⁻²	Non-covered area
Photo-absorber thickness optimisation				
1000	20 s	600 nm	~ -8.8 mA cm ⁻²	1.64 V vs. RHE
2200	20 s	365 nm	~ -11.15 mA cm ⁻²	1.77 V vs. RHE
4000	20 s	200 nm	~ -3 mA cm ⁻²	1.75 V vs. RHE
ETL thickness optimisation				
1000	30 s	55 nm	~ -7.8 mA cm ⁻²	No difference in the onset value
2000	30 s	40 nm	~ -11.15 mA cm ⁻²	
3000	30 s	34 nm	~ -9.2 mA cm ⁻²	
O-BP thickness optimisation at 0.6 V vs. RHE determining the mass activity (A g⁻¹)				
Loading amount		Thickness		

50 mg	~ 92 μm	1.924 mA cm^{-2}	0.67 A g^{-1}
100 mg	~ 206 μm	1.964 mA cm^{-2}	0.34 A g^{-1}
150 mg	~ 302 μm	2.005 mA cm^{-2}	0.23 A g^{-1}

Revised Fig. 3 | Characterisation and ORR activity of the O-BP-integrated PSK photocathode. **a,b**, Top-view and cross-sectional scanning electron micrographs of the formed MAPbI_3 PSK showing the thickness of the individual layers. **c**, J - E responses of the O-BP electrocatalyst, Field's metal (FM)/PSK photocathode, and integrated O-BP/FM/PSK photocathode device, exhibiting an anodic shift. Integrated O-BP/FM/PSK photocathode devices were simulated under 1-sun and an air mass of 1.5 G, with the averaged cyclic voltammograms (Avg. CV) scans under light and no irradiation (red arrow indicates the scan direction during the experiment) in the presence of an O_2 supply. **d**, Wavelength-dependent spectral response (i.e., incident-photon-to-current efficiency (IPCE)) and the corresponding integrated photocurrent density of the integrated device at 0 V vs. RHE. **e**, J - E response of the α - NiFeO_x/CP anode without stirring. Inset depicts the exact overpotential for the OER. **f**, Chronopotentiometry test of the α - NiFeO_x/CP anode conducted in a three-electrode set-up. **c-f** Experiments conducted in a 0.1 M KOH solution (pH ~13.17) under ambient conditions with a scan rate of 5 mV s^{-1} . All scale bars equal 1 μm .

Revised Fig. 4 | Unassisted solar H_2O_2 production with integrated photocathode system. | a, Operating point, obtained from the overlap of the $a\text{-NiFeO}_x/\text{CP}$ anode and the integrated system of the O-BP/FM/PSK photocathode, for the generated solar H_2O_2 affords a photocurrent of $\sim 2.51 \text{ mA cm}^{-2}$ for generating solar H_2O_2 in the catholyte. **b,** In a two-electrode set-up under simulated 1-sun conditions, the unassisted H_2O_2 production (n) of our integrated O-BP/FM/PSK device could continuously produce $\sim 442 \pm 15.2 \mu\text{mol cm}^{-2}$ of H_2O_2 for 12 h, without any degradation, at 0 V. **c,** Graph showing the generated rate of solar H_2O_2 on the left y-axis, while the corresponding solar-to-chemical conversion efficiency (SCC; %) has been marked on the right y-axis (Supplementary Table 2). The highest SCC efficiency peaked after 2 h of reaction, also demonstrated in Supplementary Fig. 28.

While there are several ways to determine the thicknesses of the thin-films [*Thin Solid Films* **113**, 101, (1984); *ACS Energy Lett.*, **5**, 2580, (2020)], we used the scanning electron microscopy images collected for analysing the cross-sectional view.

We have added the name of the equipment used in our work in the manuscript text.

Added contents

Manuscript Text, Page number 26, line number 500

⇒ ... which was also used to determine the thickness of the all-solution processed layer.

Manuscript Text, Page number 9, line number 172

⇒ The thickness of O-BP did not affect its performance or stability if the value was greater than ~100 μm , as shown in Supplementary Fig. 11 and Supplementary Fig. 12.

Manuscript Text, Page number 12, line number 214

⇒ We optimised the thicknesses of the individual layers, including the HTL, PSK photo-absorber, and ETL, to enhance the performance of the O-BP/FM/PSK photocathode, as shown in Supplementary Fig. 17 and Supplementary Table 1.

Comment 2: Fig. 1. How does the electron, hole, and mass (e.g., proton) transport in the whole system? A clear pathway should be provided to help the reader better understand the catalysis process.

Response: We thank the reviewer for the very important and detailed comment. We have added a plausible pathway for the transport of the electron and hole in Revised Fig.1.

Added contents:

Manuscript Text, Page number 16, line number 304

⇒ The arrows shown in Fig. 1 demonstrate the movement of the photogenerated holes (blue) and electrons (red).

Revised Fig. 1 | Schematic illustration of the system demonstrating unassisted solar hydrogen peroxide (H_2O_2) production. An inverse-structure methylammonium lead triiodide (MAPbI_3) perovskite (PSK) photocathode with oxidised buckypaper (O-BP) and an amorphous-based nickel–iron oxide ($\alpha\text{-NiFeO}_x$) with a carbon paper substrate are connected for bias-free H_2O_2 production. Embedded copper wires connect the photocathode and anode to enable the photogenerated hole–electron pairs to flow in a closed circuit. Dissolved O_2 is reduced to H_2O_2 by photocathode-mounted O-BP. The oxygen evolution reaction (OER) occurs at the $\alpha\text{-NiFeO}_x$. Definitions: PCBM, [6,6]-phenyl C_{61} butyric acid methyl ester; PEIE, polyethyleneimine; and FTO, fluorine-doped tin oxide. The red-coloured arrow shows the direction of the movement of the electrons in the illustration while the simultaneously generated holes' direction is depicted by the blue-coloured arrow.

Comment 3: What are the electrolytes for photocathode and anode respectively? 0.1 M KOH for both? The authors did not clearly mention the relevant important information in the Experiment section. I thought the experiments were conducted in alkaline media, where the H_2O_2 is in the form of a peroxide ion. How about the performance in neutral or acid media?

Response: We thank the reviewer for the very important comment. We used a 0.1 M KOH electrolyte in both the anode and cathode parts. We added this detailed information in the experimental section part.

According to reviewer's comment, we have performed linear sweep voltammetry (LSV) and cyclic voltammetry (CV) measurements in acidic, neutral, and alkaline media. As shown in the added Supplementary Fig. 18a,b, the active catalyst, i.e., O-CNT/CP, showed negative shifts in its onset value by ~ 300 mV and ~ 582 mV, whereas the integrated photocathode attained similar shifts of ~ 600 mV and ~ 900 mV in the neutral and acidic media, respectively, compared to its shifts in the 0.1 M KOH electrolyte. As a result, the intersection point plotted between the integrated photocathode and the α -NiFeO_x/CP anode showed no overlap when the reaction-set-up was carried out in the former two electrolytes, as shown in Added Supplementary Fig. 18c. This led to the zero-photocurrent density revealing that the system could not operate under bias-free conditions when the photocathode with a carbon-based active catalyst was tested in lower pH ($< \text{pH} \sim 7$) electrolytes (Added Supplementary Fig. 18d).

Added Supplementary Fig. 18 | Performance measurement of O-BP/FM/PSK photocathode device in acid, neutral, and alkaline media. **a**, Linear sweep voltammetry (LSV) measurements of the O-CNTs evaluated at a 5 mV s^{-1} scan rate. **b**, $J-E$ response of the integrated O-BP/FM/PSK photocathode device in acidic (black colour), neutral (red colour), and alkaline (blue colour) media. Integrated O-BP/FM/PSK photocathode devices were simulated under 1-sun illumination and an air mass of 1.5 G in the presence of an O_2 supply. **c**, Operating point from the overlap of the $\alpha\text{-NiFeO}_x$ /carbon paper (CP) anode and average cyclic voltammogram scan of the integrated O-BP/FM/PSK photocathode in acidic (pH ~ 1), neutral (pH ~ 7), and alkaline (pH ~ 13) media, respectively. **d**, Chronoamperometry test of O-BP/FM/PSK at 0 V vs. counter electrode.

Modified: Manuscript Text, Page number 24, line number 463

“This was separated by a Nafion 117 membrane, which was activated by maintaining in deionised water for 2 h before the reaction.”

⇒ The two-compartment reactor containing a 0.1 M KOH solution (7.5 mL) in both the anode and cathode chambers was separated by a Nafion 117 membrane, which was activated by maintaining it in deionised water for 2 h before the reaction.

Added Contents:

Manuscript Text, Page number 12, line number 214

⇒ In addition, we found that the basic conditions (0.1 M KOH, pH ~ 13) with an optimised pH provided better performance than those of the neutral (0.05 M Na_2SO_4 , pH ~ 7) and acidic (0.05 M H_2SO_4 , pH ~ 1) conditions (Supplementary Fig. 18).

Comment 4: Following the above question, unlike the electrolyte-free H_2O_2 production system, how can the produced H_2O_2 be separated from the electrolyte for practical use?

Response: We thank the reviewer for the critical and important comments. To figure out the possibility of H_2O_2 separation from the electrolyte we performed vacuum distillation [US patent, US6592840B, (2003)]. Even though H_2O_2 boils at $\sim 150^\circ\text{C}$, it is not appropriate to increase the temperature too much because H_2O_2 is chemically unstable at high temperature and decomposes. We only needed to vaporize water and H_2O_2 at mild temperatures. Here, we applied low external pressure, which helped to vaporize H_2O_2 . Review only Fig. 1 shows the photograph of the vacuum distillation instrument set-up. The external pressure was approximately 0.1 atm and the

temperature for distillation was 50 °C. To demonstrate feasibility with high concentrations of H₂O₂, we used prepared solutions containing 0.1 M KCl + 10 wt% H₂O₂ (we did low-pressure distillation after neutralizing the KOH solution with HCl, because the KOH solution would become more basic during this distillation process).

We conducted ICP-OES and a titration method (equations 1 and 2) to check the concentrations of potassium and chloride ions, respectively, before and after the low-pressure distillation (Review only Fig. 2). Review only Fig. 3. clearly shows a dramatically reduced concentration of potassium and chloride ions in the distilled samples, which demonstrates that this low-pressure distillation technique is effective in separating H₂O₂.

Review only Fig. 1 | Photograph of vacuum distillation instrument set-up for the separation of H₂O and H₂O₂ from an inorganic solution containing potassium and chloride ions.

Review only Fig. 2 | Photograph showing the colour of each sample during the titration method for determining chloride ion concentration. After consuming all chloride ions, K_2CrO_4 could be generated, simultaneously making the yellowish colour disappear and a red colour appear.

Review only Fig. 3 | The concentration of potassium (K^+) and chloride (Cl^-) ions measured by ICP-OES and a titration method, respectively, before and after the distillation process.

Comment 5: The author claimed the perovskite photocathode was very stable, by analysing the current within a given test period. More characterizations (e.g., morphology and chemical states) should be conducted to further support this point.

Response: We thank the reviewer for their valuable and significant comments. We first conducted stability measurements of the optimized O-BP/FM/PSK photocathode (according to reviewer question 1) under solar simulated light at 0 vs. counter electrode and found that it was stable for more than 45 h (Revised Supplementary Fig. 25) (Previously, we conducted the stability measurements for 20 h.) To further confirm its stability, we conducted XRD, SEM, and XPS analyses before and after the stability tests, according to the reviewers' comment (Revised Supplementary Fig. 25 and Added Supplementary Fig. 26). We found that the visual morphology, crystal structure, and electronic distribution were maintained after the 45 h of stability tests.

Revised Supplementary Fig. 25 | Unassisted solar H₂O₂ generation at 0 V vs. counter electrode. Long-term stability tests of our integrated O-BP/FM/PSK photocathode under short-circuit conditions. Experiments were conducted in a 0.1 M KOH solution (pH ~13.17) under ambient conditions with a continuous O₂ supply.

Added Supplementary Fig. 26 | Characterisation of the integrated O-BP/FM/PSK photocathode after measuring the long-term stability test. a, XRD patterns of the carbon materials before and after stability tests. **b,c,** SEM images of the O-BP before (**b**) and after (**c**) the long-term stability tests showing no accumulation of the incoming electrons at the interface rather than the rapid consumption towards the conversion into H_2O_2 . **d, e,** XPS surveys of the carbon materials before and after stability tests. All scale bars equal $1 \mu\text{m}$.

Modified: Manuscript Text, Page number 17, line number 316

“Moreover, the performance of the O-BP/FM/PSK photocathode did not decline during 20h of testing, with fluctuations in the obtained photocurrent density after 12 h, mainly due to the addition of the electrolyte (Supplementary Fig. 17).”

⇒ Moreover, the performance of the O-BP/FM/PSK photocathode did not show any sign of decrease during the 45 h of stability tests, as shown in Supplementary Fig. 25. XRD, scanning electron microscopy (SEM), and XPS analyses of the O-BP/FM/PSK photocathode after the stability tests (Supplementary Fig. 26) showed no significant changes in its crystal structure, morphology, or electronic distribution.

Comment 6: Line 250, what is the SCC after 1h reaction? It seems the FE% slightly declined after 6 hours, so the SCC% might also drop after the long time test.

Response: We thank the reviewer very much for their important and insightful comments. For the optimized O-BP/FM/PSK photocathode, by optimizing the thickness of its HTL, PSK layer, ETL and O-BP layer, it showed more stable performance than its un-optimized version. To gather more accurate stability data, we added error bars that were determined by performing the stability test several times (see error bars in Revised Fig. 4b). In addition, for accurate presentation, we wrote and plotted the SCC values after 1 h, 2 h, 6 h and 12 h stability tests in the Revised Supplementary Table 2 and Added Supplementary Fig. 28. The slight decrease in SCC at 12 h can be due to H₂O₂ decomposition at high H₂O₂ concentrations, not from the degradation of the PSK photoelectrode; this is proposed because we observed that the net photocurrent of O-BP/FM/PSK photocathode was maintained for more than 45 h without any change in its chemical composition or structure, as shown in the Revised Supplementary Fig. 25 and Added Supplementary Fig. 26.

Revised Fig. 4 | Unassisted solar H₂O₂ production with integrated photocathode system. | a, Operating point, obtained from the overlap of the *a*-NiFeO_x/CP anode and the integrated system of the O-BP/FM/PSK photocathode, for the generated solar H₂O₂ affords a photocurrent of ~2.51 mA cm⁻² for generating solar H₂O₂ in the catholyte. b, In a two-electrode set-up under simulated 1-sun conditions, the unassisted H₂O₂ production (*n*) of our integrated O-BP/FM/PSK device could continuously produce ~442 ± 15.2 μmol cm⁻² of H₂O₂ for 12 h, without any degradation, at 0 V. c, Graph showing the generated rate of solar H₂O₂ on the left y-axis, while the corresponding solar-to-chemical conversion efficiency (SCC; %) has been marked on the right y-axis (Supplementary Table 2). The highest SCC efficiency peaked after 2 h of reaction, also demonstrated in Supplementary Fig. 28.

Added Supplementary Fig. 28 | Solar-to-chemical conversion (SCC; %) graph during 12 h of reaction time. The graph demonstrates two different values. The black circles represent the mean value along with their deviation, while the highest value attained by the best performing O-BP/FM/PSK photocathode was 1.463 % (after 2 h of reaction span) which is depicted by the red stars.

Added contents:

Manuscript Text, Page number 17, line number 326

⇒ ~1.463 % after 2 h (average SCC of 1.35 ± 0.073 % after 6 h, from equation 2)

Comment 7: Supplementary Table. The electrolyte for all systems should be provided to better compare the performance. Also, there is a Typo. The year number in the last row was not correct.

Response: We thank the reviewer again for pointing out this typo error. In the Revised Supplementary Table 2, we have added the electrolyte for a better comparison of the performances with the pH values included. Also, for the convenience of the readers, we have included the reaction time after which the generation rate of H₂O₂ was determined, with the durability of the system highlighted in the table below.

Supplementary Table 2 | Solar-to-chemical conversion efficiency (SCC; %) and generation rates of unassisted solar H₂O₂ reported for photoelectrodes over the past five years. All the values were determined under unbiased conditions, i.e., 0 V vs. counter electrode applied to the two-electrode set-up

Year	Process @ 0 V vs CE	Photoelectrodes involved	Electrolyte	Generation rate $\mu\text{mol cm}^{-2} \text{min}^{-1}$	Durability time (h)	SCC (%)	Reference
2016	ORR	WO ₃ Co ^{II} (ch) on CP (generation on cathode only, 1h reaction time)	0.1 M HClO ₄ + 0.1 M NaClO ₄ , pH 1.3	0.14 0.336	24	0.273 0.655	1
2016	ORR	WO ₃ /BiVO ₄ Au (generation on-cathode only, 20 min reaction time)	2 M KHCO ₃	0.039 0.0614	-	0.076 0.12	2
2018	Dual	BiVO ₄ Carbon (400 s reaction time)	2 M KHCO ₃ , and 1 M Na ₂ SO ₄ , respectively	0.48	0.1 5	0.936	3
2020 2019	ORR	TiO ₂ Co-N-CNT* (generation on cathode only, 6 h reaction time)	0.1 M phosphate borate buffer, pH 4.5	0.021 0.035		0.040 0.069	4

2019	ORR	WO ₃ Co ^{II} (ch)	HClO ₄ (pH 1.3) + 0.1 M NaClO ₄	0.13		0.25		5
2020	ORR	BiVO ₄ pTTh (utilization of both photoelectrodes)	0.1 M KOH, pH ~12.9	0.21 0.238	14	0.409 0.464		6
2020	Dual	P-Mo-BiVO ₄ AQ-CNT/C (5 h reaction time)	1 M NaHCO ₃ , pH ~7.8	0.16	5 -	0.312		7
2020	Dual	Mo-BiVO ₄ AQ-CNT/C (5 h reaction time)	1 M NaHCO ₃ , pH ~7.8	0.11	5 -	0.214		7
2021	ORR	TiO ₂ AQ-Graphite (generation on cathode only, 100 h reaction time)	1 M H ₂ SO ₄ and 1 M KOH, respectively	0.256	100	0.499		8
2021- 2022	ORR	a -NiFeOx O-BP/FM/PSK (generation on photocathode only, 12 h reaction time)	0.1 M KOH	0.510 Avg. 0.637 0.664 0.695 0.615	12 Best 1 2 6 12	0.996 Avg. 1.19 1.29 1.35 1.2	Best 1.33 1.463 1.42 1.24	This work This work

~~*Calculation of the SCC only considers the area of the cathode~~

Dual = water oxidation and oxygen reduction simultaneously

Reaction time = duration for which the H₂O₂ generation rate was determined

Durability time = the given system was tested at 0 V in two-electrode set-up

References

- 1 Mase, K., Yoneda, M., Yamada, Y. & Fukuzumi, S. Seawater usable for production and consumption of hydrogen peroxide as a solar fuel. *Nat Commun* **7**, 11470, (2016).
- 2 Fuku, K. *et al.* Photoelectrochemical Hydrogen Peroxide Production from Water on a WO₃/BiVO₄ Photoanode and from O₂ on an Au Cathode Without External Bias. *Chem Asian J* **12**, 1111-1119, (2017).
- 3 Shi, X. J., Zhang, Y. R., Siahrostami, S. & Zheng, X. L. Light-Driven BiVO₄-C Fuel Cell with Simultaneous Production of H₂O₂. *Adv. Energy Mater.* **8**, (2018).
- 4 Ko, M. *et al.* Unassisted solar lignin valorisation using a compartmented photo-electro-biochemical cell. *Nat. Commun.* **10**, 5123, (2019).
- 5 Liu, J., Zou, Y., Jin, B., Zhang, K. & Park, J. H. Hydrogen Peroxide Production from Solar Water Oxidation. *ACS Energy Lett* **4**, 3018-3027, (2019).
- 6 Fan, W. *et al.* Efficient hydrogen peroxide synthesis by metal-free polyterthiophene via photoelectrocatalytic dioxygen reduction. *Energy Environ. Sci.* **13**, 238-245, (2020).
- 7 Jeon, T. H., Kim, H., Kim, H.-i. & Choi, W. Highly durable photoelectrochemical H₂O₂ production via dual photoanode and cathode processes under solar simulating and external bias-free conditions. *Energy Environ. Sci.* **13**, 1730-1742, (2020).
- 8 Jeon, T. H. *et al.* Solar photoelectrochemical synthesis of electrolyte-free H₂O₂ aqueous solution without needing electrical bias and H₂. *Energy Environ. Sci.* **14**, 3110-3119, (2021).

Additional Revisions

We changed some figures and statements in the main manuscript due to some minor mistakes.

- ⇒ In the entire manuscript, the abbreviated words O-CNT and CNT were replaced with O-CNTs and CNTs as their plural form and updated the verb form as appropriately needed.
- ⇒ Moreover, as per the guidelines, we have also placed the amount of substance used correspondingly to the chemical inside the parenthesis in the methods section specifically.
- ⇒ The abbreviations mentioned for the first time in the main manuscript and supplementary texts have now been acknowledged everywhere with their appropriate full names including in the captions.
- ⇒ Revised Supplementary Fig. 3: we changed Fig. d line colour from red to green.

Previous: Manuscript Text, Page number 17, line number 297

Modified: Manuscript Text, Page number 21, line number 387

dispersed for 2 h in an ultrasonic bath followed by tip sonication with 25% amplitude for 15 min
dispersed by tip sonication with a 25 % amplitude for 15 min followed by bath ultrasonication for
2 h.

Modified Supplementary Fig. 6, Page number 8

~~531.7 (532.1), 533.4 (533.4) eV, and 535.5 (535.3) eV~~
532.1 (532.0) eV, 533.4 (533.4) eV, and 535.5 (536.0) eV

REVIEWERS' COMMENTS

Reviewer #1 (Remarks to the Author):

The authors have thoroughly addressed my comments and also those of the other reviewers. I believe that the quality of the revised manuscript is very good and could be accepted without change for publication.

Reviewer #2 (Remarks to the Author):

The authors have properly addressed my comments. I am ok for its acceptance.

Reviewer #3 (Remarks to the Author):

Most comments have been well addressed. Happy to recommend a publication.